# Actor-Free Continuous Control via Structurally Maximizable Q-Functions

**Yigit Korkmaz**[1][*]    **Urvi Bhuwania**[1][*]    **Ayush Jain**[1,2][†]    **Erdem Bıyık**[1][†]

[1]Thomas Lord Department of Computer Science, University of Southern California, [2]Meta AI [‡]

## Abstract

Value-based algorithms are a cornerstone of off-policy reinforcement learning due to their simplicity and training stability. However, their use has traditionally been restricted to discrete action spaces, as they rely on estimating Q-values for individual state-action pairs. In continuous action spaces, evaluating the Q-value over the entire action space becomes computationally infeasible. To address this, actor-critic methods are typically employed, where a critic is trained on off-policy data to estimate Q-values, and an actor is trained to maximize the critic's output. Despite their popularity, these methods often suffer from instability during training. In this work, we propose a purely value-based framework for continuous control that revisits structural maximization of Q-functions, introducing a set of key architectural and algorithmic choices to enable efficient and stable learning. We evaluate the proposed actor-free Q-learning approach on a range of standard simulation tasks, demonstrating performance and sample-efficiency on par with state-of-the-art baselines, without the cost of learning a separate actor. Particularly, in environments with constrained action spaces, where the value functions are typically non-smooth, our method with structural maximization outperforms traditional actor-critic methods with gradient-based maximization. We have released our code at https://github.com/USC-Lira/Q3C.

## 1 Introduction

Reinforcement learning (RL) has been highly effective in many domains, ranging from robotics to gaming and recommender systems [25, 29, 45, 1]. Value-based RL methods such as Deep Q-Networks (DQNs) [34, 17] are widely used in discrete action spaces due to their simplicity, training stability, and sample-efficiency. However, these methods require maximization over the action space, which is feasible only in discrete action spaces, requiring alternative approaches for continuous control.

In problems with continuous action spaces, policy-based methods are typically used, such as DDPG [30], SAC [16], and TD3 [11]. These approaches employ an actor-critic architecture, where the actor (policy) is updated based on feedback from the critic (Q-function) to select the optimal actions, and both components are usually parameterized by neural networks. While powerful, actor-critic methods often face instability due to the coupled training of actor and critic, hyperparameter sensitivity, and predicting in high-dimensional action spaces. Moreover, gradient-based actor-critic methods struggle when the action space is constrained, such as when actions require to be within specific safety bounds [8, 24], because the gradient-ascending actor can only find locally optimal actions.

Can we develop an actor-free value-based algorithm that can efficiently select optimal actions in continuous domains? In this work, we propose to extend the DQN framework to continuous action spaces via a Q-function representation that is structurally maximizable. This has been explored for

---

[*]Equal Contribution,   [†]Equal Advising,   [‡] Work performed in personal capacity outside of work at Meta.

39th Conference on Neural Information Processing Systems (NeurIPS 2025).

quadratic models of the Q-function [15], but that has limited representation capacity. In contrast, our approach builds upon a prior technique that uses a set of "control-points" to anchor Q-function approximation [5, 12], such that the maximum is at one of the control-points. This direction was largely abandoned [51] due to poor benchmark performance and scalability in complex environments. We revisit this direction with a series of critical design innovations that enable practical and effective continuous Q-learning for the first time with function approximation of control-points.

We propose the Q3C algorithm, Q-learning for continuous control with control-points, where our contributions include: (1) combining a structurally maximizable Q-function with deep learning to find maxima easily in complex continuous action spaces; (2) an improved model architecture that simplifies optimization by separating the complexity of control-point generation and value estimation; and (3) algorithmic improvements that help make the algorithm robust across different environments with various action spaces and reward scales. We evaluate our approach on a range of continuous control Gymnasium tasks [50] where Q3C is on par with the performance of common deterministic actor-critic methods. Particularly, in complex environments with constrained action spaces, where gradient-based actor-critic methods struggle, Q3C consistently outperforms existing approaches. We also conduct ablation studies to measure and visualize the impact of each design component in Q3C.

## 2 Related Work

**Off-policy actor-critic methods** are widely employed for tasks with continuous action spaces. Among them, Deep Deterministic Policy Gradient (DDPG) [30] is particularly influential. As a deep variant of the deterministic policy gradient algorithm [44], DDPG combines a Q-function estimator with a deterministic actor that seeks to maximize the estimated Q-function. However, the interaction between the actor network and the Q-function estimator often introduces instability, making DDPG highly sensitive to hyperparameters and difficult to stabilize.

Various improvements to address these shortcomings have been proposed, including multi-step returns [19], prioritized replay buffers [40], and distributional critics [6]. One notable extension, Twin Delayed Deep Deterministic Policy Gradient (TD3) [11], introduces several key modifications, such as delayed updates for the policy, target networks, and the use of clipped double Q-learning, to significantly improve the stability and robustness of training, making TD3 one of the most widely adopted algorithms for continuous control. However, TD3 often performs suboptimally in environments with complex or non-convex Q-function landscapes. To address this, Jain et al. [24] propose augmenting TD3 with a successive actor framework that trains multiple actors to explore different modes of the Q-function. However, this approach introduces additional computational overhead and inference-time latency. Furthermore, the expressiveness of the actor's parameterization may still be insufficient to capture complex optimal action distributions. Finally, Soft Actor-Critic (SAC) [16] learns a stochastic policy that maximizes an entropy-regularized Q-function.

**Evolutionary algorithms** like simulated annealing [27], genetic algorithms [47], tabu search [14], and cross-entropy methods (CEM) [9] are deployed for global optimization in RL but often suffer from premature convergence at local optima in environments with complex Q-functions [20]. CEM approaches such as QT-Opt [25, 28, 26], GRAC [43], CEM-RL [35], and Cross-Entropy Guided Policies [46] are relatively effective but additionally introduce a high computational workload, struggling with high-dimensional action spaces as sampling becomes progressively more inefficient [53].

**Value-based methods**, as opposed to actor-critic methods, suffer from the challenge of finding the maximal action in continuous domains. Prior work approach this problem via various optimization techniques to minimize the Bellman residual [4], including mixed-integer programming [39], the cross-entropy method [25], and gradient ascent [31]. While these approaches can be effective, they often result in either suboptimal local maximization or are computationally impractical [24]. Another line of work discretizes the action space and performs Q-weighted averaging over discrete actions [32]. Though effective in low-dimensional settings, this approach does not scale well to high-dimensional tasks. Gu et al. [15], Wang et al. [52] propose a different approach by constructing the state-action advantage function in quadratic form, allowing for analytical solutions to directly obtain the Q-value maximum. However, this formulation restricts the Q-function's expressivity to be quadratic in action space, restricting its practical applicability. Amos et al. [2] make a similarly limiting assumption that the Q-function is universally convex in actions and use a convex solver for action selection. Most similar to our approach, Asadi et al. [3] proposed RBF-DQN, which learns

Q-functions without an actor by using a deep network with a radial basis function (RBF) output layer, enabling easy identification of the maximum action-value. However, standard RBF interpolation does not guarantee that the maximum lies at a basis centroid. It has an implicit smoothing trade-off between how expressive the Q-function can be in modeling various local optima and how close a basis centroid is to the true maxima. In contrast, the control-point approximation framework of Q3C alleviates this trade-off, being notably more adaptable to arbitrarily shaped Q-functions, including those with multiple modes or non-convexities. Thus, Q3C exhibits the ability to find the accurate maximizing action, which was the primary bottleneck of prior value-based methods.

Our work builds on a less known but promising framework that approximates Q-values using control-points, also referred to as **wire-fitting interpolators** [5, 12, 13]. While earlier studies included this approach as a baseline, they reported poor or unstable performance [51, 31]. In this work, we revisit this framework and propose several novel modifications that are crucial to stabilize learning, enabled by advances in deep learning and reinforcement learning. We demonstrate that, with these critical additions, control-point-based Q-function approximators become competitive with state-of-the-art reinforcement learning algorithms in standard benchmarking environments and outperform them on constrained action space environments.

# 3 Preliminaries

## 3.1 Q-Learning

We consider a Markov Decision Process (MDP), where at each time step $t$, an agent observes the state $s_t \in \mathcal{S}$ and takes an action $a_t \in \mathcal{A}$ in the environment $\mathcal{E}$. The agent receives a reward $r(s_t, a_t)$, and transitions to the next state $s_{t+1} \in S$. The objective of the agent is to learn a deterministic policy $\pi : \mathcal{S} \to \mathcal{A}$ that maximizes the cumulative return discounted by a factor of $\gamma \in [0, 1)$ per time-step, $R_t = \sum_{i=t}^{\infty} \gamma^{(i-t)} r(s_i, a_i)$. Q-learning [48, 33] solves this by defining the optimal action-value function $Q^*(s, a)$ as the maximum expected return achievable after taking action $a$ at state $s$,

$$Q^*(s, a) = \max_{\pi} \mathbb{E}\left[R_t \mid s_t = s, a_t = a, \pi\right].$$

The greedy policy is optimal, $\pi^* = \arg\max_{a \in \mathcal{A}} Q^*(s, a)$, where $Q^*$ follows the Bellman equation,

$$Q^*(s, a) = \mathbb{E}_{s' \sim \mathcal{E}}\left[r + \gamma \max_{a' \in \mathcal{A}} Q^*(s', a') \mid s, a\right]. \tag{1}$$

Value iteration algorithms like Deep Q-Networks (DQN) [33] apply the Bellman equation as an iterative update to train a function approximation $Q$ with weights $\theta$, called the Q-network,

$$L_{\text{Bellman}}(\theta) = \mathbb{E}_{s, a \sim \rho}\left[\left(\left(r + \gamma \max_{a' \in \mathcal{A}} Q(s', a'; \theta) - Q(s, a; \theta)\right)^2\right)\right], \tag{2}$$

where $\rho(s, a)$ is the probability distribution over states and actions in the collected training data.

## 3.2 Maximization in Q-Learning

The two $\max$ operations involved in deriving the greedy policy from Eq. (1) and evaluating the next state's best Q-value in Eq. (2) are easily computable in discrete action spaces by evaluating the Q-values of every action [33, 18, 34, 17]. However, they become infeasible in continuous action spaces. To address this, the deterministic policy gradient algorithm [44, 30, 11] learns an "actor" policy in addition to the "critic" Q-function. The actor is trained with gradient ascent of the Q-function landscape, resulting in a greedy policy that finds locally optimal actions. Furthermore, the $\max$ in Eq. (2) is avoided by replacing the max-Q formulation with the expected-Q of the actor [48, 44, 30].

However, the introduction of an additional actor brings several downsides: (i) additional hyperparameters for the actor network, making reproducibility a challenge due to interaction between the learning of the actor and critic [21], (ii) computational overhead of increased training memory, and (iii) actors trained with the policy-gradient can only find locally optimal actions [24].

# 4 Q3C: Q-learning for Continuous Control with Control-points

We aim to simplify Q-learning in continuous action spaces by proposing a lightweight actor-free approach that enables the maximization required in Equations (1) and (2). Our key insight is to learn a structurally maximizable representation of the Q-network that alleviates the need for an actor. Previously, Baird and Klopf [5] introduced *wire-fitting*, a general function approximation system that uses a finite number of control-points for fitting the Q-function. This design reduces the problem of finding the maximum over the entire action space to finding the maximum among the control-points. However, the direct application of wire-fitting to deep neural networks has led to poor results [51, 31]. We identify the crucial challenges that have limited *deep* wire-fitting and propose our approach, Q-learning for continuous control with control-points (Q3C), to address them.

## 4.1 Function Approximation Using Control-points

The wire-fitting framework [5] facilitates finding the maximum of a function $f : \mathcal{U} \to \mathbb{R}$ by maintaining a set of $N$ control-points $\mathcal{U}_c = \{u_1, \ldots u_N\}$ and their corresponding $f$ values $\mathcal{Y}_c = \{y_1 \ldots y_N\}$. The value $f(u)$ is evaluated as inverse weighted interpolation smoothed with $c_i \geq 0$,

$$f(u) = \frac{\sum_i y_i w_i}{\sum_i w_i}, \text{ where } w_i = \frac{1}{|u - u_i|^2 + c_i (y_{\max} - y_i)}. \tag{3}$$

As shown in Figure 1, this formulation guarantees that the maximum of the function $f$ is attained at one of the control-points, $u_i \in \mathcal{U}_c$ such that $i = \arg\max_k y_k$.

The smoothing parameter in our formulation affects the function's value only at non-maximal control-points. With small smoothing, the approximated function passes through all control-points, whereas with a large smoothing parameter, the function's value may differ slightly at some control-points. However, in all cases, it is still guaranteed that the maximum of the function lies at a control-point—an advantage in highly non-convex Q landscapes.

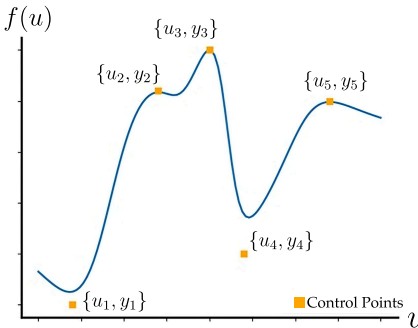

Figure 1: **Wire-fitting function.** A Q-function represented by an interpolation of learnable control-points is structurally maximized at one of the control-points.

This function approximation is extended to model Q-functions $Q(s, a)$ with the goal of finding the maximizing action $a$ for any given state $s$. Herein, the control-points and their values are learned as functions of $s$: $\mathcal{A}_c(s) = \{\hat{a}_1(s), \ldots \hat{a}_N(s)\}$ and $\mathcal{Q}_c(s) = \{\hat{Q}_1(s), \ldots \hat{Q}_N(s)\}$. These sets of functions $\mathcal{A}_c$ and $\mathcal{Q}_c$ are modeled with function approximations like neural networks, for example, a shared neural network backbone with $2N$ scalar output heads for $\hat{a}_i(s)$ and $\hat{Q}_i(s)$.

$$Q(s, a) = \frac{\sum_i \hat{Q}_i(s) \, w_i(s, a)}{\sum_i w_i(s, a)}, \text{ where } w_i(s, a) = \frac{1}{|a - \hat{a}_i(s)|^2 + c_i \, \max_k \left(\hat{Q}_{\max} - \hat{Q}_i(s)\right)}. \tag{4}$$

This representation of the Q-function enables Q-learning in continuous action spaces with the ability to find the maximal action via a direct maximization over scalars $\hat{Q}_i(s)$,

$$\arg\max_a Q(s, a) = \hat{a}_j \text{ where } j = \arg\max_i \hat{Q}_i. \tag{5}$$

In Appendix B, we prove the following proposition that replacing a neural network Q-function with wire-fitting interpolation in the Q-function preserves its universal approximation ability.

**Proposition.** *Let $\mathcal{A}$ be a compact action set and $s$ be a given state. For any continuous Q-function $Q_s(a) := Q(s, a)$ and any $\epsilon > 0$, there exists a finite set of control-points $\{\hat{a}_1, \ldots, \hat{a}_N\} \subset \mathcal{A}$ with corresponding values $y_i = Q_s(a_i)$ such that the wire-fitting interpolator*

$$f(a) = \frac{\sum_{i=1}^N y_i \, w_i(a)}{\sum_{i=1}^N w_i(a)}, \qquad w_i(a) = \frac{1}{|a - a_i| + \max_k(y_k - y_i)}, \text{ satisfies}$$

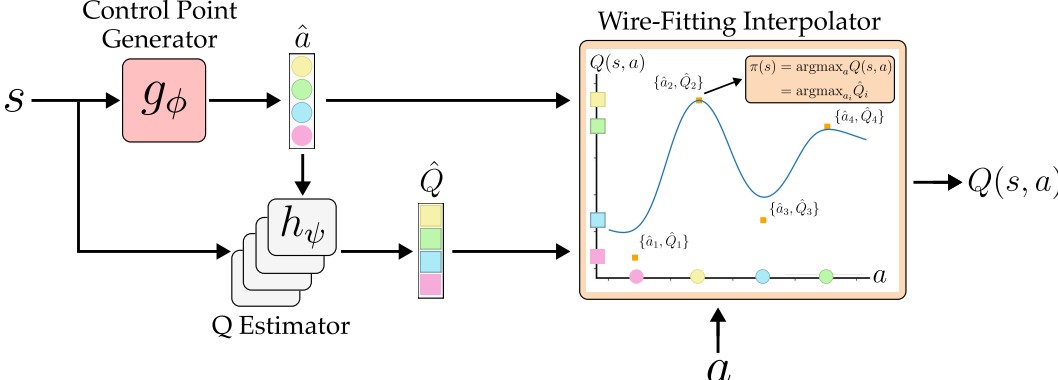

Figure 2: **Q3C Architecture** consists of 3 components: (i) a control-point generator estimates the representative $N$ control-point actions, (ii) a Q-estimator estimates the values of these $N$ points, and (iii) a wire-fitting interpolator estimates the inverse-distance weighted Q-value of the given action.

$$\|f - Q_s\|_\infty = \sup_{a \in \mathcal{A}} |f(a) - Q_s(a)| < \epsilon.$$

*Hence, the wire-fitting formulation can approximate any continuous $Q(s, \cdot)$ arbitrarily well.*

**Extension to Deep Reinforcement Learning.** Prior attempts on adapting the above wire-fitting framework to deep neural networks struggled on high-dimensional continuous control [31]. This is in line with prior works in deep RL [33, 30] that show the need for specialized stabilization techniques in the RL algorithm and architecture. Therefore, we propose to (i) build on the exploration and function approximation tools of the TD3 algorithm [11] which analogously learns deterministic policies for continuous control, (ii) reduce optimization complexity of the control-point architecture (Section 4.2), and (iii) improve the robustness of training across different environments (Section 4.3).

**Building on TD3.** We explore with Gaussian-noise added over the deterministic action selected by greedy maximization. We augment the training with twin Q-networks to avoid overestimation and target networks to make the learning targets stationary in Eq. (2). To encourage generalization, we apply target policy smoothing to the maximizing action $a' \in \mathcal{A}$, obtained via Eq. (5).

## 4.2 Reducing Optimization Complexity of Control-points

While building on TD3 provides the basic framework for deep wire-fitting, the control-point architecture still suffers from two complexities of optimization: (1) a control-point's Q-value $\hat{Q}_i(s)$ is learned without conditioning on its corresponding action $\hat{a}_i(s)$ and must implicitly learn it in neural weights, and (2) Q-function's global expressivity with many control-points comes at the cost of local learning inefficiency because the backpropagated gradient is spread over all the control-points.

**Action-conditioned Q-value Generation.** Independently predicting the Q-values $\hat{Q}_i(s)$ from their control-points $\hat{a}_i(s)$ lets the network assign very different values to identical or near-identical actions, destabilizing learning. We instead structure the control-point architecture into 2 stages: a control-point generator $g_\phi(s)$ outputs $N$ control-point actions $\hat{a}_i(s)$, for which a separate Q-estimator obtains the corresponding Q-values, $\hat{Q}_i(s) = h_\psi(s, \hat{a}_i)$. The idea of evaluating various control-points with the same Q-estimator is motivated from prior work on action representation generalization [22, 23], and ensures consistency of Q-values and simplifies learning (see Figure 2).

**Relevance-based Control-point Filtering.** To enforce stricter locality than the soft weighting in Eq. (3), we evaluate $Q(s, a)$ only using the top $k$ control-point weights $w_i(s, a)$ for the action $a$ discarding the remaining $N - k$. This hard filter removes spurious influence from distant points, sharpens the local value landscape, and yields a simpler learning task. Empirically, choosing $k \ll N$ consistently improves both training stability and final performance across benchmarks.

---

**Algorithm 1** Q3C

---

Initialize control-point generators $g_{\phi_1}, g_{\phi_2}$ and Q estimators $h_{\psi_1}, h_{\psi_2}$
Initialize target networks $\psi_i', \phi_i' \leftarrow \psi_i, \phi_i$
Initialize replay buffer
Define `calc_q_value(s, a, φ, ψ)` $\leftarrow$ following Eq. (4)
Define `get_action(s)` $\leftarrow$ Eq (5)
**for** $t = 1$ to $T$ **do**
    **Select Action**
    Select action with exploration noise and observe reward $r$ and new state $s'$.
    Store transition tuple $(s, a, r, s')$ in replay buffer
    **Update**
    Sample mini-batch of $N$ transitions $(s, a, r, s')$ from replay buffer
    $\tilde{a} \leftarrow$ `get_action`$(s') + \epsilon$, where $\epsilon \sim \text{clip}(\mathcal{N}(0, \tilde{\sigma}), -\delta, +\delta)$
    $Q_i(s', \tilde{a}) \leftarrow$ `calc_q_value`$(s', \tilde{a}, \phi_i', \psi_i')$ for $i = 1, 2$
    $y \leftarrow r + \gamma \min_{i \in \{1,2\}} Q_i(s', \tilde{a})$
    $Q_i(s, a) \leftarrow$ `calc_q_value`$(s, a, \phi_i, \psi_i)$ for $i = 1, 2$
    $L_{\text{Bellman}}(\phi_i, \psi_i) \leftarrow N^{-1} \sum (y - Q_i(s, a))^2$,    $L_{\text{seperation}}(\phi_i) \leftarrow$ Eq. (6)
    Calculate losses $L(\phi_i, \psi_i) = L_{\text{Bellman}}(\phi_i, \psi_i) + L_{\text{seperation}}(\phi_i)$ for $i = 1, 2$
    Update parameters $\phi_i, \psi_i$ for $i = 1, 2$
    **if** $t \mod d = 0$ **then**
        Update target networks:
        $\phi_i' \leftarrow \tau\phi_i + (1 - \tau)\phi_i'$
        $\psi_i' \leftarrow \tau\psi_i + (1 - \tau)\psi_i'$
    **end if**
**end for**

---

## 4.3 Robustness of Training across Different Tasks

The performance of the wire-fitting framework may still degrade when reward scales, action ranges, or environment dynamics differ markedly across tasks. To be a robust algorithm, Q3C must (i) maintain a diverse, well-spread set of control-points so the critic models a representative slice of the action space in every state, and (ii) normalize the scales of the action distance and Q-value difference despite varying reward magnitudes across states and tasks.

**Control-point Diversity.** While action-conditioned Q-value generation disentangles Q-values from control-point generation, it does not ensure that control-points cover the action space. In practice, we observe that control-points often cluster near the boundaries, limiting the expressiveness of the learned Q-function (see Figure 6). While policies acting at extremes can perform well in certain scenarios [42], more uniformly distributed control-points offer richer representations and improved robustness when applying Q3C. To spread them out, we add a pairwise *separation* loss with $\varepsilon \ll 1$:

$$L_{\text{separation}}(\phi) = \frac{1}{N(N-1)} \sum_{i \neq j} \frac{1}{\|\hat{a}_i(s) - \hat{a}_j(s)\|_2 + \varepsilon}, \text{ where } N = \# \text{ of control-points} \quad (6)$$

which is minimized when the points are uniformly dispersed. We add this loss to the Bellman loss in Eq. (2) weighted by a hyperparameter, $\lambda \in (0, 1]$.

**Scale-Aware Control-points and Q-values** We normalize action spaces to $[-1, 1]$ and use a $tanh$ nonlinearity in the control-point generator. In Eq. (4), while the action distances $|a - \hat{a}_i(s)|^2$ are bounded in $[0, 1]$, the Q-value difference term $c_i \cdot (y_{\max} - y_i)$ can vary widely between environments, even with shared $c_i = c$. We (i) rescale each state's control-point values to $\tilde{Q}_i \in [0, 1]$ within the weight term only, $\tilde{Q}_i = \frac{\hat{Q}_i - \hat{Q}_{\min}}{\hat{Q}_{\max} - \hat{Q}_{\min}}$ and (ii) anneal the smoothing factor $c_i$ exponentially. This prevents large rewards from overwhelming spatial information and keeps learning robust across diverse tasks. Combining all the aforementioned modifications, we present Q3C, a pure value-based reinforcement learning algorithm for continuous control in Algorithm 1.

Table 1: **Standard Environments.** Final performance on Classic Mujoco Environments shows that Q3C is comparable to TD3 while outperforming other baselines (mean ± std).

| Environment | TD3 | NAF | Wire-Fitting | RBF-DQN | Q3C |
|---|---|---|---|---|---|
| Pendulum-v1 | $-144.64 \pm 25.28$ | $-252.36 \pm 56.63$ | $-351.52 \pm 390.18$ | $-143.88 \pm 23.86$ | $-159.53 \pm 16.46$ |
| Swimmer-v4 | $300.70 \pm 125.64$ | $20.63 \pm 12.62$ | $313.63 \pm 106.23$ | $92.38 \pm 44.97$ | $316.40 \pm 14.75$ |
| Hopper-v4 | $3113.41 \pm 888.17$ | $500.80 \pm 240.93$ | $1987.50 \pm 1127.06$ | $2189.37 \pm 1093.13$ | $3206.14 \pm 407.23$ |
| BipedalWalker-v3 | $309.62 \pm 10.94$ | $-108.19 \pm 34.76$ | $70.01 \pm 100.28$ | $265.35 \pm 74.38$ | $290.11 \pm 26.43$ |
| Walker2d-v4 | $4770.82 \pm 560.16$ | $2179.56 \pm 1034.59$ | $2462.30 \pm 1095.41$ | $781.58 \pm 282.66$ | $3977.39 \pm 879.70$ |
| HalfCheetah-v4 | $9984.74 \pm 1076.58$ | $3531.50 \pm 802.84$ | $7546.23 \pm 1234.31$ | $6175.57 \pm 3044.93$ | $9468.66 \pm 949.01$ |
| Ant-v4 | $5167.68 \pm 673.44$ | $-18.10 \pm 0.30$ | $1154.59 \pm 420.92$ | $1674.03 \pm 964.60$ | $3698.41 \pm 1314.88$ |

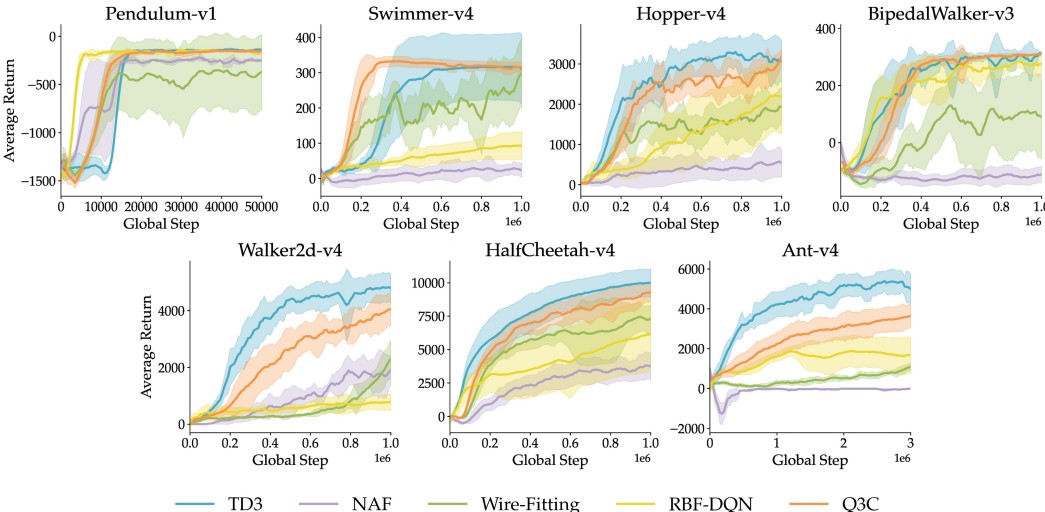

Figure 3: **Standard Environments.** Q3C is comparable in performance to TD3 and outperforms other actor-free value-based baselines in most environments.

## 5   Experiments

We conduct experiments across a range of continuous control tasks and baselines. Our goal is to evaluate both performance in standard benchmark environments and robustness in settings with complex, multimodal Q-functions. Further details about the environments can be found in Appendix E.

**Environments.** We evaluate our method on several tasks from the Gymnasium suite [50]. Specifically, we use Pendulum, Swimmer, Hopper, Bipedal Walker, Walker2d, Half Cheetah, and Ant tasks to cover a range of task difficulty. In all experiments, we use state-based observations and do not modify the reward function of the tasks. In addition, similar to prior work [24, 39], we create restricted versions of a subset of the environments, namely Inverted Pendulum, Hopper, and HalfCheetah. We follow the same procedure as Jain et al. [24], where the action space is restricted by defining a set of hyperspheres as the valid section of the space. The actions outside this space are defined as invalid and have no effect on the environment. These restricted settings are designed to induce non-convex Q-functions, where local maximization fails to find the optimal action.

**Baselines.** We primarily compare Q3C against deterministic RL algorithms, including both actor-critic and value-based methods, as our approach results in a deterministic policy and follows TD3's exploration. Whenever possible, we use the official implementations from the stable-baselines3 [37] and tuned hyperparameters from rlzoo3 [36] to ensure optimized baselines. We implement Q3C using the same stable-baselines3 backbone to minimize implementation-level differences between methods.

**Q3C (Ours)**: Q-learning for continuous control that learns a structurally maximizable Q-function via a set of learned control-points with improved optimization and robustness for deep networks.

**Wire-Fitting** [12, 13]: Vanilla wire-fitting Q-function approximation without our contributions.

Table 2: **Restricted Environments.** Q3C outperforms TD3 and actor-free baselines (mean ± std).

| Environment | TD3 | NAF | Wire-Fitting | RBF-DQN | Q3C |
|---|---|---|---|---|---|
| InvertedPendulumBox | $782.76 \pm 348.92$ | $909.72 \pm 120.14$ | $386.38 \pm 307.57$ | $862.02 \pm 398.05$ | $1000 \pm 0$ |
| HalfCheetahBox | $2276.70 \pm 2036.59$ | $4867.05 \pm 1487.69$ | $-2139.78 \pm 4702.25$ | $2238.38 \pm 3227.31$ | $4357.82 \pm 1503.33$ |
| HopperBox | $1406.83 \pm 1162.72$ | $461.54 \pm 389.04$ | $169.78 \pm 812.22$ | $1641.15 \pm 796.76$ | $1974.28 \pm 1170.05$ |

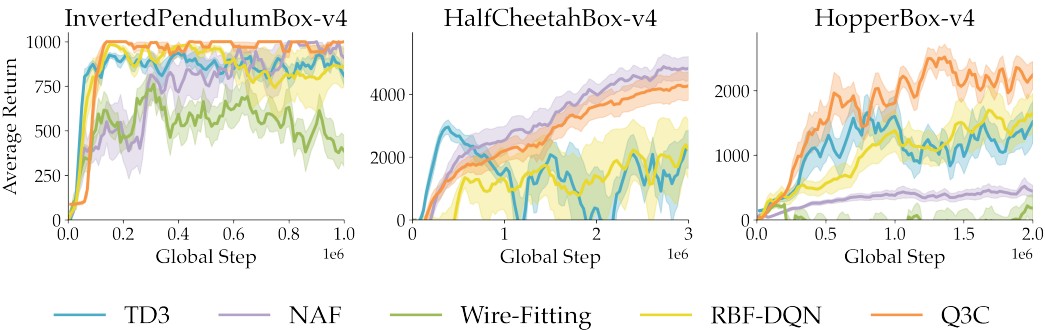

Figure 4: **Restricted Environments.** Q3C consistently outperforms TD3, RBF-DQN, and Wire-Fitting in restricted environments and largely outperforms NAF. Q3C converges at a higher reward with high stability while other algorithms are stuck at local optima.

**TD3** [11]: A state-of-the-art actor-critic algorithm that improves stability in continuous control by using double Q-learning, delayed policy updates, and target policy smoothing.

**NAF** [15]: A value-based algorithm for continuous action spaces that assumes the Q-function to a quadratic form, allowing the optimal action to be computed analytically without an explicit actor.

**RBF-DQN** [3]: A deep Q-learning algorithm variant that uses radial basis functions to approximate Q-functions in continuous action spaces.

**Evaluation Scheme.** We train 10 different random seeds for each algorithm. Throughout training, we evaluate each method every 10000 steps by running 10 rollout episodes and report the average return. The curves correspond to the mean, and the shaded region to one standard error across 10 trials.

## 5.1 Standard Environments

We present our quantitative results in Table 1 and learning curves in Figure 3. Q3C achieves performance comparable to TD3 on most tasks, with the exception of Ant-v4, where it performs suboptimally. Compared to the vanilla wire-fitting baseline, which lacks our proposed additions, Q3C achieves substantial improvements across all benchmarks, highlighting the impact of its components.

Other value-based algorithms than Q3C struggle in most environments. NAF consistently underperforms, likely due to its restrictive inductive bias that the Q-function is quadratic in the action space, which does not hold for complex control problems. RBF-DQN similarly achieves suboptimal performance, possibly because its function approximation cannot reliably recover the true maximizing action without excessive smoothing. Furthermore, it requires a large number of centroids ($\sim 100$) to achieve sufficient Q-function expressivity, limiting scalability to high-dimensional environments.

These results show that Q3C is a viable alternative to deterministic actor-critic methods in common RL benchmarks and the state-of-the-art actor-free method for continuous action spaces.

## 5.2 Restricted Environments

Environments with restricted action spaces help evaluate robustness of Q3C to complex Q-value landscapes [24]. These constraints induce sharp discontinuities in the Q-values of nearby actions, because they result in significantly different outcomes and returns. As a consequence, the Q-function becomes highly non-convex and difficult to optimize. Our experiments on restricted environments presented in Figure 4 and Table 2 show that TD3 performs significantly worse in

Table 3: **Ablations of Q3C.** Final performance comparison shows every component is necessary to improve the converged performance of Q3C (mean $\pm$ std).

| Algorithm | Hopper-v4 | BipedalWalker-v3 | Walker2d-v4 | HalfCheetah-v4 |
|---|---|---|---|---|
| Q3C | **3206.14 $\pm$ 407.23** | **290.11 $\pm$ 26.43** | **3977.39 $\pm$ 879.70** | **9468.66 $\pm$ 949.01** |
| Q3C - CondQ | 2329.9 $\pm$ 610.1 | 286.0 $\pm$ 46.7 | 3614.3 $\pm$ 488.7 | 8385.9 $\pm$ 564.9 |
| Q3C - Ranking | 3036.5 $\pm$ 371.4 | 179.8 $\pm$ 187.3 | 3167.5 $\pm$ 1167.2 | 8960.9 $\pm$ 519.6 |
| Q3C - Div | 1921.1 $\pm$ 1117.9 | -67.8 $\pm$ 119.6 | 3174.3 $\pm$ 1194.7 | 5282.7 $\pm$ 1116.2 |
| Q3C - Norm | 2915.3 $\pm$ 366.6 | 261.5 $\pm$ 83.4 | 2880.1 $\pm$ 1327.5 | 8745.5 $\pm$ 529.4 |
| Wire-Fitting | 1987.50 $\pm$ 1127.06 | 70.01 $\pm$ 100.28 | 2462.30 $\pm$ 1095.4 | 7546.23 $\pm$ 1234.31 |

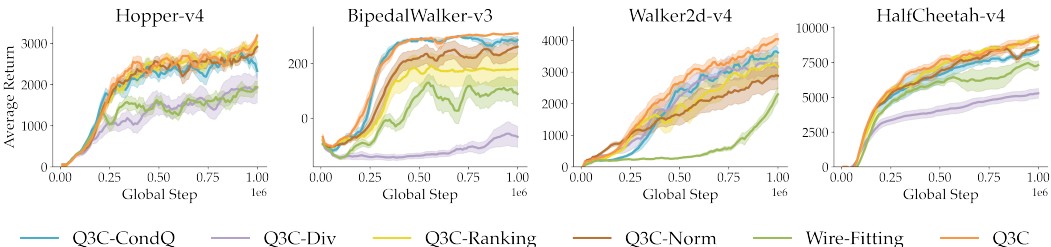

Figure 5: **Ablations of Q3C.** Every component of Q3C is validated for importance. Individual Q3C components complement each other and combined Q3C model visibly outperforms ablations.

restricted environments than in their unrestricted counterparts. This is expected since gradient-based optimization of the policy can get stuck at local optima in highly non-convex Q-functions. In contrast, Q3C consistently outperforms TD3 by avoiding restrictive policy parameterizations and instead leveraging direct maximization over a learned Q-function.

Q3C similarly outperforms other value-based algorithms. The vanilla wire-fitting algorithm achieves minimal to no performance in the restricted environments, highlighting the necessity of our additional design components. RBF-DQN and NAF achieve moderate performance but are still suboptimal. This is probably due to similar drawbacks to those outlined in Section 5.1 for standard environments, but are exacerbated with the increased complexity of restricted environments. NAF performs marginally better than Q3C in HalfCheetahBox-v4, suggesting that an approximation with a quadratic function might be enough to sufficiently represent the Q-function in this environment.

Overall, these results highlight Q3C's ability to handle discontinuities in the action space. We also present extended comparisons against other popular RL algorithms in Appendix D.2.

# 6 Ablations

To understand the contribution of each component of our method, we conduct ablation experiments by selectively removing one component at a time from Q3C. The ablated components include:

- Q3C without conditional Q-value generation (Q3C-CondQ)
- Q3C without control-point diversification (Q3C-Div)
- Q3C without ranking of relevant control-points (Q3C-Ranking)
- Q3C without normalization of Q-value differences in the wire-fitting kernel (Q3C-Norm)

We evaluate these variants on four environments—Hopper, BipedalWalker, Walker2d, and HalfCheetah—and report learning curves in Figure 5, and the corresponding final reward values in Table 3. For reference we also include the vanilla wire-fitting approach in the plots. Further ablations on the design choices can be found in Appendix C.

Across all tasks, full Q3C outperforms all ablated variants, highlighting the importance of each component. While the impact of each feature varies by environment, removing any single component

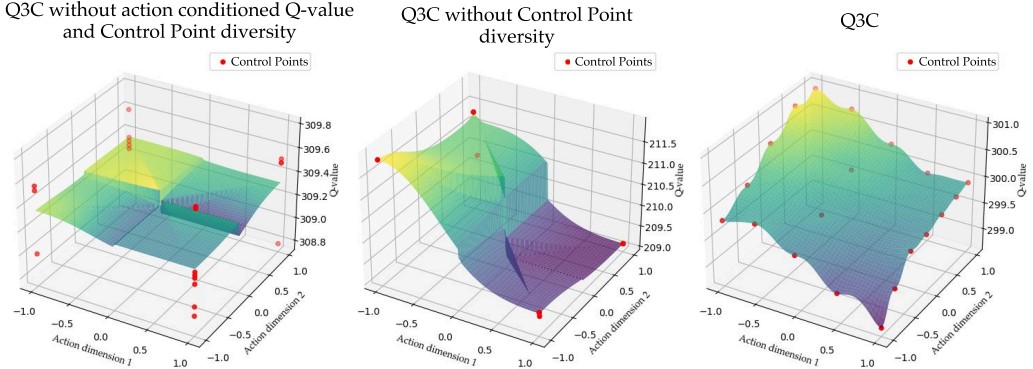

Figure 6: **Learned Q-Function Visualizations of Key Q3C Component Ablations.** Conditional Q-Value Architecture and control-point Diversity Loss are Essential For Reasonable Q-Functions

generally results in performance and stability degradation. Notably, even the weakest ablated variants outperform vanilla wire-fitting, demonstrating the effectiveness of our contributions.

One exception is the Q3C-Div variant: in certain environments, removing the control-point diversification loss leads to a substantial performance drop. We attribute this to the underlying Q-value landscapes of those environments, and the role of diversification in ensuring sufficient action space coverage, which is essential for the other components—such as relevance filtering and Q-value normalization—to operate effectively.

## 6.1 Visualizing the Effect of Q3C Components

As seen in Figure 6, without explicit constraints on control-points (left), the learned control-points suffer from stacking, regression to corner action spaces, and noticeably minimal distribution (Section 4.1). With the architectural modification to condition the Q-values on the control-points (center), the control-points still stack towards the corners, but all control-points with the same action vector now retain the same Q-value, which is a key improvement in consistency (Section 4.2). When extending this revised architecture with control-point diversity (Section 4.3), Q3C achieves both explicit consistency across all control-points and a solid control-point distribution that prevents premature collapsing at endpoints and encourages expressivity.

## 7 Conclusion

In this work, we introduced Q3C, an actor-free Q-Learning approach for off-policy reinforcement learning in continuous action spaces. Our approach builds on the control-point function approximator, augmented with several key architectural innovations, such as action-conditioned Q-value generation, relevance-based control-point filtering, encouraging control-point diversity, and normalized scaling for robustness. We demonstrated that these additions enable the function approximator to match or exceed the performance of state-of-the-art online RL algorithms on standard benchmarks. Moreover, in constrained environments where only a subset of the action space is admissible, Q3C achieves robust performance while TD3's gradient maximization often fails.

**Limitations and Future Work.** Q3C simply adopts TD3's exploration scheme, and its sample efficiency can lag behind other baselines in certain environments. More work on exploration strategies could be promising, such as Boltzmann over the control-point values. As a hybrid between DQN-style methods and actor-critic algorithms, Q3C offers flexibility to incorporate enhancements from both paradigms. Future work could explore integrating sample-efficiency improvements such as $n$-step returns [18], prioritized experience replay [40], or using batchnorm layers in the critic network instead of target networks [7]. Additionally, adapting the control-point approximator to the offline RL setting could be an exciting direction: due to its inherent constraints on Q-value interpolation, it may offer natural mitigation against overestimation, a common challenge in offline learning. Finally, we only employ Q3C on deterministic Q-learning, however, future work can investigate extending it to stochastic policies where the control-point architecture models a soft-Q function like SAC.

## Acknowledgments

We thank the USC Center for Advanced Research Computing (CARC) for providing us with compute resources. Urvi Bhuwania was supported by a CURVE fellowship from the USC Viterbi School of Engineering. This project was partially funded by Airbus Institute for Engineering Research (AIER). We appreciate the valuable discussions with anonymous reviewers and members of the USC Lira Lab. We thank Norio Kosaka for starter code of restricted environments.

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

# A Impact Statement

This paper introduces a purely value-based reinforcement learning framework for continuous control, aimed at improving both stability and tractability in complex action spaces. By avoiding restrictive policy parameterizations and enabling direct action selection through a structured Q-function, our method is particularly well-suited for tasks with constrained or safety-critical action spaces. While our approach improves robustness and computational efficiency over existing baselines, it does not explicitly account for fairness, interpretability, or real-world deployment risks. Care must be taken when applying this method in safety-sensitive or high-stakes domains, where additional mechanisms—such as constraint-aware training or human oversight—may be necessary to ensure reliable and ethical deployment.

# B Proof for Proposition

We show that replacing a neural network Q-function with wire-fitting interpolation in the Q-function preserves its universal approximation ability.

**Proposition.** *Let $\mathcal{A}$ be a compact action set and $s$ be a given state. For any continuous Q-function $Q_s(a) := Q(s, a)$ and any $\epsilon > 0$, there exists a finite set of control-points $\{\hat{a}_1, \ldots, \hat{a}_N\} \subset \mathcal{A}$ with corresponding values $y_i = Q_s(a_i)$ such that the wire-fitting interpolator*

$$f(a) = \frac{\sum_{i=1}^{N} y_i \, w_i(a)}{\sum_{i=1}^{N} w_i(a)}, \qquad w_i(a) = \frac{1}{|a - a_i| + c_i \max_k(y_k - y_i)}, \text{ satisfies}$$

$$\|f - Q_s\|_\infty = \sup_{a \in \mathcal{A}} |f(a) - Q_s(a)| < \epsilon.$$

*Hence, the wire-fitting formulation can approximate any continuous $Q(s, \cdot)$ arbitrarily well.*

*Proof.* We prove this proposition by following the classical convergence analysis of inverse–distance interpolation [10], which was adapted to the control–point formulation for reinforcement learning by Baird and Klopf [5].

**Uniform–continuity radius.** Because $\mathcal{A}$ is compact and $Q_s$ is continuous, $Q_s$ is uniformly continuous; hence there exists $\delta > 0$ such that $|Q_s(a) - Q_s(b)| < \epsilon/2$ whenever $|a - b| < \delta$.

**Control-point $\delta$–net.** Choose a finite $\delta$–net $\{a_i\}_{i=1}^{N} \subset \mathcal{A}$, i.e. for every $a \in \mathcal{A}$ there is an index $i^\star(a)$ with $|a - a_{i^\star}| < \delta$. Define the targets $y_i := Q_s(a_i)$ and the positive offsets $c_i > 0$ (a constant choice $c_i \equiv 1$ suffices).

**Interpolant as a convex combination.** For each $a \in \mathcal{A}$ set

$$w_i(a) = \frac{1}{|a - a_i| + c_i \, \Delta_i}, \qquad \Delta_i := \max_k (y_k - y_i), \quad \lambda_i(a) := \frac{w_i(a)}{\sum_k w_k(a)}.$$

Because all $w_i(a) > 0$, the $\lambda_i(a)$ form a partition of unity and the wire–fitting interpolant is $f(a) = \sum_i \lambda_i(a) \, y_i$.

**Error decomposition.** Write $f(a) - Q_s(a) = E_{\text{near}} + E_{\text{far}}$, where

$$E_{\text{near}} = \sum_{i: |a - a_i| < \delta} \lambda_i(a) \big(Q_s(a_i) - Q_s(a)\big),$$

$$E_{\text{far}} = \sum_{i: |a - a_i| \geq \delta} \lambda_i(a) \big(Q_s(a_i) - Q_s(a)\big).$$

*Near indices.* For the "near" set the uniform–continuity condition yields $|Q_s(a_i) - Q_s(a)| < \epsilon/2$; hence $|E_{\text{near}}| \leq \epsilon/2$.

*Far indices.* Let $R := \max_{\mathcal{A}} Q_s - \min_{\mathcal{A}} Q_s$ and $c_{\min} := \min_i c_i$. For any $j$ with $|a - a_j| \geq \delta$,

$$\frac{w_j(a)}{w_{i^\star}(a)} \leq 1 + \frac{c_{\min} R}{\delta} =: \theta.$$

It follows that $\sum_{j:|a-a_j|\geq\delta}\lambda_j(a)\leq\theta^{-1}$. Choosing $\delta < c_{\min}R\,\epsilon/(2R-\epsilon)$ makes $\theta^{-1} < \epsilon/(2R)$ so that $|E_{\text{far}}| < R\,\theta^{-1} < \epsilon/2$.

**Uniform bound.** Combining the two parts gives $|f(a) - Q_s(a)| < \epsilon$ for every $a \in \mathcal{A}$; hence $\|f - Q_s\|_\infty < \epsilon$. □

## C  Implementation Details and Ablations for Q3C

### C.1  Control-point Diversification Functions

In addition to the separation loss we defined in Eq. (6), we experiment with an alternative loss formulation that explicitly encourages each control-point to maintain a minimum distance from its nearest neighbor:

$$\mathcal{L}_{\text{min\_N}}(\phi) = \frac{1}{N}\sum_{i=1}^{N}\min_{\substack{1\leq j\leq N \\ j\neq i}}\|x_i - x_j\|_2$$

This variant is used for a subset of tasks—specifically, Hopper and BipedalWalker—where we empirically observed improved performance. These similar objectives aim to enhance the expressivity and robustness of the Q-function approximation by avoiding clustered or redundant control-points.

### C.2  Smoothing Coefficient

We experimented with three primary methods of setting the smoothing parameter $c$: tuning the constant value of smoothing as a hyperparameter, learning the smoothing coefficient, and scheduling a decay for smoothing.

We find in Figure 7 that an exponentially decaying smoothing scheduler typically works better than learning a smoothing parameter. Utilizing a scheduler induces a greater stability in the smoothing parameter that allows the model to learn the Q-function and distribution of control-points more effectively. The learning process becomes less susceptible to erratic shifts in the smoothing parameter.

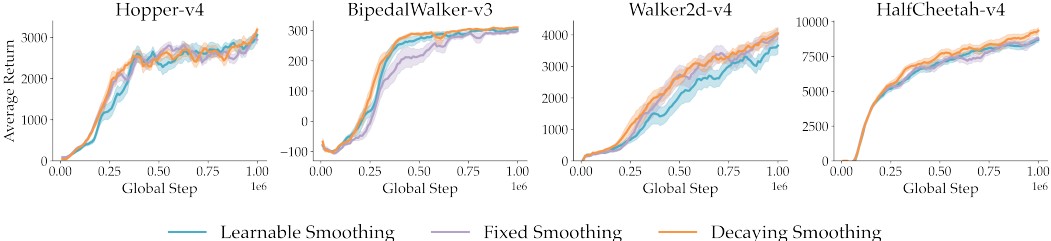

Figure 7: Ablations of Q3C with different strategies for smoothing parameter: The method of decaying smoothing is consistently more stable as justified in section C.2.

### C.3  Learning Rate Schedulers

In traditional actor-critic architectures like TD3, there is often an in-built policy delay with the critic being updated more frequently than the actor. However, since Q3C employs the same network to serve as both actor and critic, our implicit policy is updated at the same rate as our critic. Thus, Q3C is more sensitive to various learning schemes to stabilize training.

We tried six different learning rate schedulers: constant, linear decay, exponential decay, inverse exponential decay, cosine decay, and one-cycle learning rate. For each scheduler except for the constant learning schedule, we kept the maximum learning rate as the learning rate specified in the hyperparameters and the minimum learning rate as 10% of the maximum learning rate.

Among these, we found that inverse exponential decay and cosine decay were the optimal learning schedulers across all environments. For sake of consistency, we apply a delayed exponential decay learning scheduler for all results with the final learning rate set to 10% of the initial learning rate.

### C.4 Control-points with K-Nearest Neighbors

We experimented across various control-point combinations of $N$ (total # of control-points) and $k$ (top $k$ rankings), and found that setting $N = 20$ and $k = 10$ consistently worked best for nearly all environments. We utilize this setting as a starting point for hyperparameter tuning, as it balances coverage and computational cost. We then progressively increase the value of $N$, keeping $k$ relatively constant, until learning is optimal. We employ $N = 30$ and $k = 15$ for Ant-v4, and $N = 70$ and $k = 10$ for Adroit environment. However, our algorithm is reasonably robust across different control-point combinations, as shown in Figure 8.

As a general rule, increasing the number of control-points $N$ allows for greater expressivity but involves learning a more complex Q-function, which may slow down learning. Due to this tradeoff, the optimal number of control-points $N$ tends to scale more conservatively than a proportional increase with action dimensionality (e.g., AdroitHandHammer-v1's action space is 26-dim but only requires 70 control-points). Increasing the value of $k$ typically leads to a smoother and continuous Q-function across the action space, whereas a smaller value of $k$ allows for more granular and piecewise control of the Q-function.

Additionally, the conditional Q architecture enables parallelization across control-points during Q-value generation, ensuring that the parameter count does not scale linearly with $N$. This parallel structure supports Q3C's scalability to high-dimensional and non-convex Q-function landscapes.

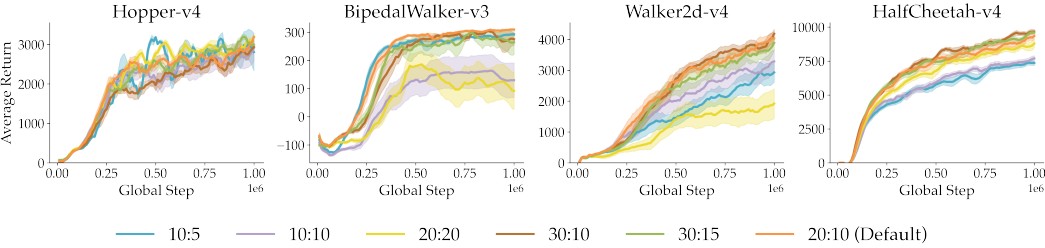

Figure 8: Ablations of Q3C with different configurations for # of control-points (N), and top k ranking. Different configurations are represented as N:k in the legend.

## D  Further Experiment Results

In addition to ablations of the core method modifications, we have provided extra results to demonstrate the effect of the specific implementation details listed above.

### D.1  Q3C's
### Performance in High-Dimensional Action Spaces

We have conducted preliminary experiments on Adroit Hand Hammer environment [38] to evaluate Q3C's performance in high-dimensional action spaces (26-dim). As seen in Figure 9, Q3C is able to match TD3, while surpassing SAC, which shows that Q3C is able to scale its learning to high-dimensional action spaces.

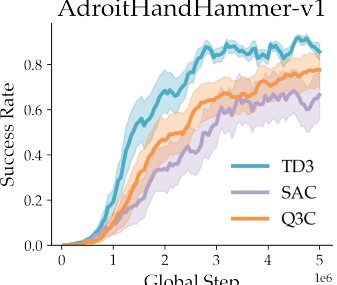

Figure 9: Success rate plot of Q3C against TD3 and SAC on High-dimensional environments.

### D.2  Comparison Against SAC & PPO

In addition to the baselines reported in the main paper, we also compare Q3C with two widely used algorithms, SAC and PPO [41, 16]. The results are shown in Figure 10 and Figure 11, alongside TD3. PPO is an on-policy method and is therefore typically less sample-efficient than off-policy approaches. SAC, on the other hand, employs a different exploration scheme based on soft Q-value maximization, which differs fundamentally from the noise-based exploration used by the baselines in the main paper. Because of these differences, SAC and PPO are not directly comparable for evaluating the specific contribution of Q3C as a "structurally maximizable Q-function". We therefore do not expect, nor

claim, that Q3C should outperform these methods across all settings. Nevertheless, Q3C achieves competitive results in all standard environments and consistently matches or surpasses state-of-the-art performance in the restricted environments.

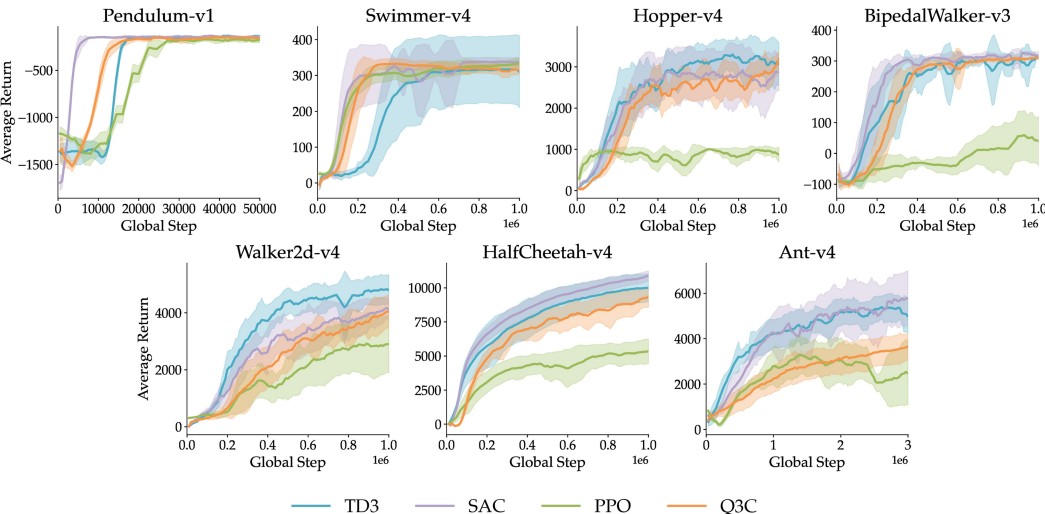

Figure 10: Comparison against TD3, SAC, and PPO on Standard Environments.

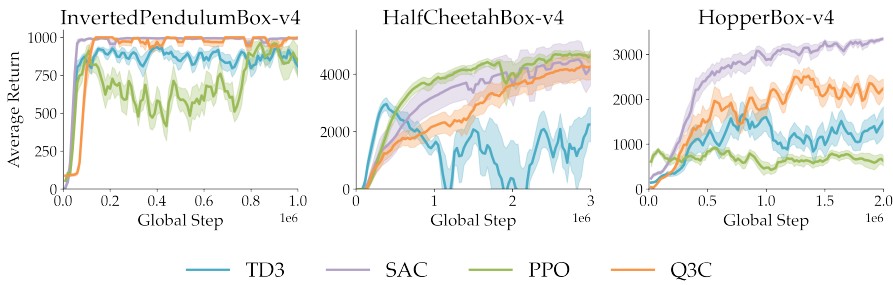

Figure 11: Comparison against TD3, SAC, and PPO on Restricted Environments.

## D.3 Wall-clock Time and Memory Footprint of Q3C

We conducted detailed comparisons of wall-clock time and memory usage between Q3C, SAC, PPO, and TD3 on a subset of evaluation environments. Experiments were run with 5 random seeds each: restricted environments on Tesla P100 GPUs and unrestricted environments on A100 GPUs. To ensure fair comparisons, we defined a reward threshold for each environment (Hopper: 3000, BipedalWalker: 300, HalfCheetah: 9000, InvertedPendulumBox: 1000, HalfCheetahBox: 4000, HopperBox: 2000) corresponding to what is generally considered optimal performance. We then recorded the wall-clock time required by each algorithm to reach and stabilize around the threshold. Algorithms marked as "N/A" did not reach or sustain the target reward within the full training duration. Results are summarized in Table 4.

Table 4: Wall-clock time comparison of different algorithms across environments.

| Algorithm | Hopper | BipedalWalker | HalfCheetah | InvertedPendulumBox | HalfCheetahBox | HopperBox |
|---|---|---|---|---|---|---|
| Q3C | 58.3 min | **44.6 min** | 61.5 min | **12.43 min** | 4 hr | **0.7 hr** |
| TD3 | **32.9 min** | 45.6 min | **39.2 min** | N/A | N/A | N/A |
| SAC | 76.3 min | 50.5 min | 69.2 min | 16.5 min | 6.6 hr | 1.3 hr |
| PPO | N/A | N/A | N/A | N/A | **1.0 hr** | N/A |
| NAF | N/A | N/A | N/A | N/A | 7.2 hr | N/A |

Table 5: Comparison of PyTorch memory allocation across algorithms.

| Algorithm | PyTorch Allocated (MB) |
|-----------|------------------------|
| Q3C | 26.58 |
| TD3 | 23.51 |
| SAC | 20.01 |
| PPO | 18.38 |

In HopperBox and InvertedPendulumBox environments, Q3C consistently achieves the target reward in significantly less time than all baselines. For instance, in InvertedPendulumBox, it reaches optimal performance in about 60% of the time required by SAC, while other baselines fail to converge. In terms of memory usage, although Q3C utilizes a parallelized evaluation of Q-values of all the control-points, the increase in memory footprint is marginal and does not hinder training.

From a runtime perspective, Q3C removes the actor network, which reduces computational overhead. However, this advantage is partially offset by the additional cost introduced by the interpolation mechanism. Since we have used a simple interpolation module, its overhead cancels out the time savings from removing the actor. We consider this an engineering limitation rather than a conceptual one: with better parallelization and optimized implementations, Q3C could realize significant runtime advantages.

More importantly, from a memory perspective—particularly in large-scale settings—Q3C demonstrates clear advantages over TD3. We analyzed memory usage across increasing network sizes while ensuring that the critic parameter count remained equivalent between Q3C and TD3, isolating the effect of the actor. As shown in Table 6, we observe increasing memory savings for Q3C as network size scales.

Table 6: Parameter counts and memory usage of Q3C and TD3 across model scales.

| Model scale* | Q3C params | TD3 params | Q3C memory (MB) | TD3 memory (MB) | % Memory saved |
|--------------|-----------|-----------|-----------------|-----------------|----------------|
| Small | 37M | 52M | 341 | 400 | 14.8% |
| Medium | 135M | 226M | 707 | 1,057 | 33.1% |
| Large | 538M | 906M | 2,245 | 3,675 | 38.9% |
| X-Large | 1.57B | 2.65B | 6,212 | 10,348 | 40.0% |

*Mapping of Layer Sizes: Small → 1,024 control-points, Medium → 2,048, Large → 4,096, X-Large → 7,000.

## D.4 Augmenting Q3C with Cross-Q

While Q3C generally converges to competitive final performance, we observe that it occasionally lags behind other baselines in terms of sample efficiency. Recent work by Bhatt et al. [7] addresses this issue in actor-critic settings by removing the target network and introducing batch normalization in the critic (and optionally in the actor), resulting in improved sample efficiency. Motivated by this, we also tried adopting a similar modification into our framework. Specifically, we removed the target network and applied batch normalization within the Q-value generation network. We evaluated this variant on the Walker2d task. Our preliminary results indicate that this approach improves sample efficiency during the early stages of training; however, the final performance remains comparable to the original Q3C. Improving sample efficiency remains an important direction for future work. We plan to further explore architectural and algorithmic modifications to Q3C that can accelerate learning while preserving the stability and performance of the full model.

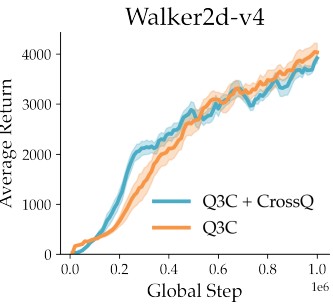

Figure 12: Comparison of Q3C with Q3C + Cross Q style modifications

# E   Environment Details

We test Q3C and our baselines across 7 Gymnasium environments [50, 49] and 4 custom restricted environments [24]. The restricted environments are variations of their corresponding standard versions where the action spaces are limited to a series of hyperspheres within the region to create complex Q-function landscapes. The traditional control environments are described in Table 7 and Figure 13.

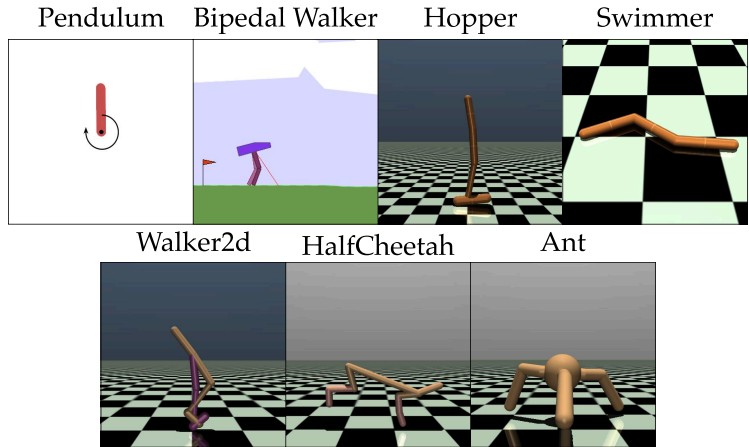

Figure 13: Visualizations of the testing environments.

Table 7: Environment Summaries with Observation and Action Spaces

| Environment | Summary | Observation / Action Space |
|---|---|---|
| Pendulum-v1 | A pendulum starts at a random angle and must be swung up and balanced upright using torque. | 3D obs / 1D cont act |
| Hopper-v4 | A one-legged robot must learn to hop forward without falling. | 11D obs / 3D cont act |
| HalfCheetah-v4 | A 2D bipedal robot with a flexible spine must learn to run forward. | 17D obs / 6D cont act |
| BipedalWalker-v4 | A two-legged robot must learn to walk across rough terrain. | 24D obs / 4D cont act |
| Swimmer-v4 | A 2D snake-like robot must swim forward in a viscous fluid. | 8D obs / 2D cont act |
| Walker2d-v4 | A 2D robot with two legs must learn to walk forward while keeping balance. | 17D obs / 6D cont act |
| Ant-v4 | A four-legged robot (quadruped) must learn to move forward stably in 3D. | 111D obs / 8D cont act |

## E.1   Restricted Environments

The restricted environments are derived directly from their standard MuJoCo counterparts. The observation and action dimensions, environment goals, and transition dynamics remain identical between the standard and restricted versions. The only modification lies in the admissible action space. However, as demonstrated in Figure 14, this modification alone is sufficient to introduce significant irregularities into the optimal Q-function.

Following prior work, the restricted action space is constructed as a union of hyperspheres embedded in the original action space. The volume of these hyperspheres is scaled by a coefficient $\kappa$, which determines the degree of restriction. A larger $\kappa$ corresponds to larger hyperspheres and thus a less constrained action space, while a smaller $\kappa$ induces tighter restrictions by shrinking the valid regions. For some environments, we lower the coefficient $\kappa$ from the original implementation to achieve an even more restricted action space, as detailed in Table 8.

Table 8: Action Hypershere Volume Multiplier

|  | InvertedPendulumBox | HopperBox | HalfCheetahBox | Walker2dBox |
|---|---|---|---|---|
| Default $\kappa$ | 3.0 | 1.65 | 1.0 | 1.0 |
| Q3C $\kappa$ | 3.0 | 1.25 | 0.25 | 0.25 |

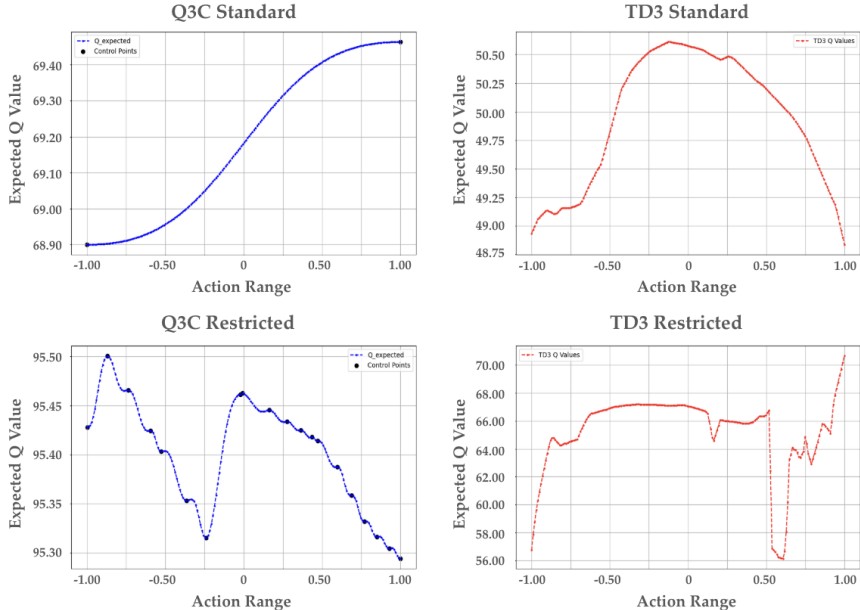

Figure 14: The Learned Q-Functions for TD3 and Q3C are notably more complex and have more local optima on the restricted version of InvertedPendulum as compared to the standard Mujoco version.

# F    Implementation Details and Hyperparameters

## F.1    Baselines

For TD3, SAC, and PPO we have used the official implementations from Stable-Baselines3[*] with the optimized hyperparameters taken from RL Zoo[*]. For NAF and RBF-DQN, we have followed the original publications and used the official codebase and hyperparameters, where available. If the hyperparameters are not available, we performed our own hyperparameter tuning. While we made a strong effort to identify competitive configurations, we acknowledge that better parameter combinations may exist.

## F.2    Q3C Hyperparameters

Q3C adopts its hyperparameters from its underlying implementation of TD3, but we tune certain important hyperparameters such as learning rate and learning starts. Furthermore, we tune the hyperparameters specific to Q3C such as the number of control-points, the number of nearest neighbors $k$, separation loss weight, and initial smoothing value.

---

[*] https://github.com/DLR-RM/stable-baselines3/blob/master/stable_baselines3
[*] https://github.com/DLR-RM/rl-baselines3-zoo/tree/master/hyperparams

### F.2.1  Classic Environments

Table 9: Q3C Hyperparameters (Part 1 of 2)

| Parameter | Pendulum | Hopper | BipedalWalker | Ant |
|---|---|---|---|---|
| Total timesteps | 1e5 | 1e6 | 1e6 | 3e6 |
| Discount factor ($\gamma$) | 0.98 | 0.99 | 0.98 | 0.99 |
| Batch size | 64 | 256 | 256 | 256 |
| Buffer size | 2e5 | 1e5 | 2e5 | 5e5 |
| Learning starts | 1000 | 30000 | 10000 | 10000 |
| Target network update ($\tau$) | 0.005 | 0.005 | 0.005 | 0.005 |
| Noise type | Gaussian | Gaussian | Gaussian | Gaussian |
| Noise std | 0.1 | 0.1 | 0.1 | 0.1 |
| Learning rate | 1e-3 | 1e-3 | 3e-4 | 5e-4 |
| Network dim | [400,300] | [400,300] | [400,300] | [400,300] |
| Target update interval | 4 | 4 | 1 | 4 |
| Gradient clipping | 10.0 | 10.0 | 1.0 | 5.0 |
| # of control-points | 3 | 20 | 20 | 30 |
| k | 3 | 10 | 10 | 15 |
| Separation loss weight | 0.01 | 0.01 | 1.0 | 0.1 |
| Initial smoothing value | 0.1 | 0.01 | 0.001 | 0.001 |

Table 10: Q3C Hyperparameters (Part 2 of 2)

| Parameter | HalfCheetah | Swimmer | Walker2d |
|---|---|---|---|
| Total timesteps | 1e6 | 1e6 | 1e6 |
| Discount factor ($\gamma$) | 0.99 | 0.9999 | 0.99 |
| Batch size | 256 | 256 | 256 |
| Buffer size | 1e5 | 5e4 | 1e5 |
| Learning starts | 30000 | 10000 | 10000 |
| Target network update ($\tau$) | 0.005 | 0.005 | 0.005 |
| Noise type | Gaussian | Gaussian | Gaussian |
| Noise std | 0.1 | 0.1 | 0.1 |
| Learning rate | 1e-3 | 1e-3 | 1e-3 |
| Network dim | [400,300] | [400,300] | [400,300] |
| Target update interval | 4 | 4 | 4 |
| Gradient clipping | 10.0 | 10.0 | 10.0 |
| # of control-points | 20 | 20 | 20 |
| k | 10 | 10 | 10 |
| Separation loss weight | 1.0 | 1.0 | 0.1 |
| Initial smoothing value | 0.01 | 0.1 | 1 |

### F.2.2 Restricted Environments

Table 11: Q3C Hyperparameters for Restricted Environments

| Parameter | InvertedPendulumBox | HopperBox | HalfCheetahBox |
|---|---|---|---|
| Total timesteps | 1e6 | 2e6 | 3e6 |
| Discount factor ($\gamma$) | 0.99 | 0.99 | 0.99 |
| Batch size | 256 | 256 | 256 |
| Buffer size | 2e5 | 1e5 | 1e6 |
| Learning starts | 1000 | 30000 | 30000 |
| Target network update ($\tau$) | 0.005 | 0.005 | 0.005 |
| Noise type | Gaussian | Gaussian | Gaussian |
| Noise std | 0.1 | 0.1 | 0.1 |
| Learning rate | 1e-3 | 1e-3 | 3e-4 |
| Network dim | [400,300] | [400,300] | [400,300] |
| Target update interval | 4 | 4 | 4 |
| Gradient clipping | 10.0 | 10.0 | 10.0 |
| # of control-points | 3 | 20 | 30 |
| k | 3 | 10 | 10 |
| Separation loss weight | 0.01 | 0.1 | 1.0 |
| Initial smoothing value | 0.1[*] | 1.0 | 0.000001[*] |

[*]Smoothing value is fixed throughout training

