# OpenReview forum: "Actor-Free Continuous Control via Structurally Maximizable Q-Functions"
_NeurIPS.cc/2025/Conference — NeurIPS 2025 poster_

### Official Review · Reviewer_mqYt · 2025-06-28

**Clarity:** 3
**Significance:** 3
**Originality:** 3
**Rating:** 5
**Confidence:** 4

**Summary:**

This manuscript introduces Q3C, which is an actor-free RL algorithm designed for continuous control tasks. The key idea is to approximate the Q-function with a set of learned control points via a wire-fitting interpolator, so that the maximum Q-value for any state is achieved at one of these actions. By turning the maximization problem into a comparison over a finite set of candidate actions, Q3C sidesteps the need for a separately trained actor to navigate the continuous action space. To make this idea suitable with deep RL, several architectural and algorithmic enhancements are proposed on top of the TD3 algorithm to stabilize training and improve performance. These enhancements include action-conditioned Q-value generation, relevance-based control-point filtering, a diversity loss to encourage control points to cover the action space, and control point scaling. Empirical evaluations across standard Gymnasium environments demonstrate that Q3C achieves performance comparable to TD3 on most tasks. However, in tasks with constrained action spaces where the valid actions are limited to certain region and therefore Q-functions tends to be highly non-convex, Q3C significantly outperforms TD3.

**Questions:**

1. Could the authors provide runtime comparisons or memory usage statistics relative to TD3 and NAF?
2. There are some notational inconsistencies throughout the manuscript. For instance, equation 4 is quite understandable but it is not clear how action-conditioned Q-value generation fits within equation 4. Do authors mean to say $Q_i(s) = h_\psi(s, a_i(s))$?
3. What is the typical number of control points that is needed? Does it have to be the same for all state-action pairs? Does it grow with action dimensionality or task complexity? Please discuss the trade-offs involved with the choice of the number of control points.
4. How robust is Q3C to the choice of the number of control points and other new hyperparameters (top-$k$, $\gamma$, etc.)? The method introduces several knobs – it would help to know if these were fixed across all experiments or tuned per environment.
5. The manuscript mentions that future work could extend Q3C to stochastic settings. Could the authors elaborate on the challenges and potential approaches for modelling soft-Q functions with control points?

**Ethical Concerns:**

["NO or VERY MINOR ethics concerns only"]

**Final Justification:**

The majority of my concerns have been addressed in the rebuttal, and the authors have conducted extensive additional evaluations. Accordingly, I have raised my score.

**Limitations:**

yes

**Quality:**

3

**Strengths And Weaknesses:**

* Strengths

1. The manuscript presents a simple yet effective solution via control-points-based wire-fitting interpolators to the long-standing “max-Q” problem in continuous action spaces. This is an interesting solution that departs from conventional actor-critic methods.
2. Empirical results demonstrate that Q3C matches the performance of the TD3 algorithm on most Gym environments, and significantly outperforms it in challenging settings with constrained action ranges. The fact that Q3C can find better policies where gradient-based actors get stuck in local optima demonstrates a clear advantage of the proposed method.  Additionally, the manuscript includes comprehensive ablations to pinpoint the contribution of each component in the design of Q3C.
3. The proposed method is also well-motivated theoretically since the wire-fitting formulation can approximate any continuous $Q(s,.)$ arbitrarily well based on proposition 3.

* Weaknesses
1. Given that the performance of Q3C is largely on par with TD3, the main benefit of is in the method being actor-free, which potentially means smaller memory footprint and faster training. However, there are no discussion on the computational complexity or experiments to evaluate the computational speed ups.

2. Q3C introduces new hyperparameters such as the number of control points and $\gamma$, in addition to the control point generator. It is not clear how different choices for these hyperparameters impact the performance of Q3C, and the manuscript does not explore how sensitive the results are to these settings

3. The evaluation could be more complete if other baselines are included such as SAC. Also, while the benchmark suite is diverse, the manuscript does not explore real-world tasks where control-point scalability could be more challenging.

---

> ### Author Rebuttal · Authors · 2025-07-31
>
> We thank the reviewer for their insightful comments. We are glad that they found our approach interesting and appreciated both the theoretical and empirical validations of our work. Below, we provide responses to the questions and clarifications requested.
>
> ## Wall-clock Time and Memory Efficiency
>
> We conducted detailed comparisons of wall-clock time and memory usage between Q3C, SAC, PPO, and TD3 in the restricted environments (5 seeds each, restricted environments on Tesla P100 and unrestricted on A100). The results are summarized below:
>
> **Wall Time until convergence**
>
> | Algorithm | Hopper | BipedalWalker | HalfCheetah | InvertedPendulumBox | HalfCheetahBox | HopperBox |
> |-----------|--------|---------------|-------------|---------------------|----------------|-----------|
> | Q3C | 58.3 min | **44.6 min** | 61.5 min | **12.43 min** | 4 hr | **0.7 hr** |
> | TD3 | **32.9 min** | 45.6 min | **39.2 min** | N/A | N/A | N/A |
> | SAC | 76.3 min | 50.5 min | 69.2 min | 16.5 min | 6.6 hr | 1.3 hr |
> | PPO | N/A | N/A | N/A | N/A | **1.0 hr** | N/A |
> | NAF | N/A | N/A | N/A | N/A | 7.2 hr | N/A |
>
>
> **Pytorch Memory**
>
> | Algorithm | Pytorch Allocated (MB) |
> |-----------|------------------------|
> | Q3C       | 26.58                  |
> | TD3       | 23.51                  |
> | SAC       | 20.01                  |
> | PPO       | 18.38                  |
>
> In HopperBox and InvertedPendulumBox environments, Q3C consistently achieves the target reward in significantly less time than all baselines. For instance, it requires approximately 60% of the time taken by SAC to reach the same performance, while other baselines often fail to reach that level at all. In terms of memory usage, although Q3C utilizes a parallelized evaluation of Q-values of all the control points, the increase in memory footprint is marginal and does not hinder training.
>
> ## New baselines: SAC, PPO, RBF-DQN
>
> As suggested by the reviewer, we have added SAC and PPO, as well as the RBF-DQN method recommended by Reviewer UQVA, as baselines. These methods were trained for the same number of timesteps as Q3C and TD3, which we also include for reference. SAC and PPO were implemented using Stable-Baselines3 (SB3) and trained with hyperparameters from RL Zoo3. For RBF-DQN, we used the official implementation and hyperparameters provided by the authors.
>
> | Environment | Q3C | TD3 | SAC | PPO | RBF-DQN |
> |-------------|-----|-----|-----|-----|---------|
> | Pendulum-v1 | -159.53 ± 16.46 | -144.64 ± 95.28 | -140.31 ± 31.14 | -176.85 ± 23.70 | 159.68 ± 15.80 |
> | Hopper-v4 | 3206.14 ± 407.23 | 3113.41 ± 888.17 | 2922.56 ± 590.18 | 882.85 ± 286.17 | 1912.34 ± 1203.35 |
> | Walker2d-v4 | 3977.39 ± 879.70 | 4770.82 ± 560.16 | 4091.98 ± 405.00 | 2917.58 ± 895.72 | 903.62 ± 308.31 |
> | HalfCheetah-v4 | 9468.66 ± 949.01 | 9984.74 ± 1076.58 | 10909.02 ± 416.07 | 5397.83 ± 746.90 | 7267.01 ± 1546.27 |
> | InvertedPendulumBox-v4 | 1000 ± 0 | 782.76 ± 348.92 | 993.40 ± 7.15 | 839.41 ± 372.97 | 734.83 ± 560.40 |
> | HalfCheetahBox-v4 | 4357.82 ± 1503.33 | 2276.70 ± 2036.59 | 4248.83 ± 2184.33 | 4637.60 ± 686.19 | 3023.26 ± 1980.03 |
> | HopperBox-v4 | 1974.28 ± 1170.05 | 1406.83 ± 1162.72 | 3330.64 ± 140.25 | 637.12 ± 449.74 | 1225.27 ± 793.74 |
>
> Q3C achieves competitive performance in all standard environments and matches or surpasses the state-of-the-art in all restricted environments. In contrast, the newly added baselines exhibit varying performance across different environments. This supports the claim that Q3C is a robust and effective alternative to actor-critic methods.
>
> Furthermore, we have conducted preliminary experiments on two environments with high-dimensional action spaces. Without any hyperparameter optimization, Q3C is able to match the final performance of TD3 in AdroitHandHammer (26-dim action space) and AdroitHandDoor (28-dim action space) environments, which shows that Q3C is able to scale its learning to high-dimensional actions.
>
> **Final Eval Success Rate on Adroit Envs**
>
> | Environment | Q3C | TD3 |
> |-------------|-----|-----|
> | AdroitHandHammer-v1 | 0.9 | 0.9 |
> | AdroitHandDoor-v1 | 0.92 | 0.84 |
>
> ## Equation 4 Clarification
>
> When conditional Q-value generation is used, the formulation in Equation 4 remains structurally the same. Specifically, the control-point actions are defined as $\hat{a}\_i = g\_{\phi}(s)\_i$ and the corresponding Q-values are given by $\hat{Q}\_i = h\_{\psi}(s, g\_{\phi}(s)\_i)$, where $g\_{\phi}$ is the control point generator network and $h\_{\psi}$ is the Q-estimator network for those control points. The full expression is provided below for completeness:
>
> $$
> Q(s,a) = \frac{\sum\_i \hat{Q}\_i(s) \,w\_i(s, a)}{\sum\_i w\_i(s, a)}
> $$
> where
> $$
> w\_i(s, a) = \frac{1}{\lvert a -  g\_\phi(s)\_i\rvert^2 + c\_i \,\max\_k\bigl(\hat Q\_{\max}- h\_\psi(s, g\_\phi(s)\_i)\bigr)}.
> $$
>
> ## Selecting number of control points and other hyperparameter ablations
>
> Regarding the hyperparameter ablations, we refer the reader to the supplementary material (Appendix D.4). We also report all of the hyperparameter configurations used in different environments in (Appendix G.3). In addition to the ablations in the main paper, we did extra ablation studies on the number of control points, top-k ranking and different smoothing strategies. Notably, we found that simply increasing the number of control points n does not always improve performance. A too large n runs the risk of overfitting or unnecessarily increasing the problem complexity which slows down convergence. A typical recipe is to start with the default n=20 and try a larger number of control points with a fixed top-k filter (k=10), which balances coverage and computational cost, leading to stable and expedited convergence.
>
> We find that the optimal number of control points tends to scale more conservatively than a proportional increase with action dimensionality due to the constant tradeoff between increasing representational potential and increasing the learning difficulty for the model. However, a limited number of additional control points is beneficial for environments with increased general task complexity and considerably higher action dimensionality. With relatively complicated environments such as Ant, HalfCheetahBox, and Adroit, we found an optimal control point combination to be between n=30,40,50 and k=10. For environments with an even higher dimensionality, we can utilize a larger n with relatively minimal increases to computational cost and problem complexity, as the backpropagation still only needs to optimize across the top k control points in a single backward pass.
>
> ## Q3C for soft Q-functions
> The basic idea to extend Q3C to stochastic settings is to replace control points with control distributions. This would require at least the following changes to the algorithm:
> 1. Instead of control points $\hat{a}\_i$, we generate control distribution mean and std: $\mu\_i, \sigma\_i$.
> 2. A Q-value is not associated with each control point distribution, and its estimator would take $s, \mu\_i, \sigma\_i$ as an input.
> 3. Finally, to evaluate the Q-value of a given action, an interpolator is still used. However, the weights of the interpolator would change from inverse-distance to the probability of the action under the control distribution: $w_i = p(a | \mathcal{N}(\mu\_i, \sigma\_i^2))$
> 4. At inference, instead of computing the $\arg\max$ over all control points, one can sample an action from the Gaussian Mixture Model (GMM) defined by the N control distributions $N(\mu\_i, \sigma\_i^2)$.
>
> ----
>
> We thank you for your insightful feedback and hope we've addressed your comments adequately. We are happy to answer any further questions and incorporate any further suggestions.

---

> > ### Comment · Reviewer_mqYt · 2025-08-04
> >
> > Thank you for responding to my previous questions. I was genuinely excited about the proposed actor-free RL algorithm, especially given its potential as a viable training method for large-scale settings under resource constraints. However, it's somewhat disappointing to see that the wall-clock time and memory footprint do not reflect a clear advantage in this regard.

---

> ### Author Response · Authors · 2025-08-05
>
> Thank you for the clarification. We now understand the reviewer’s concern, and would like to address it in detail:
> - From a runtime perspective, Q3C removes the actor network, which reduces computational overhead. However, this advantage is partially offset by the additional cost introduced by the interpolation mechanism. Since we've used a simple interpolation module, its overhead cancels out the time savings from removing the actor. In future work, we would optimize it with better parallelism and alternative implementations. That said, we believe this is an engineering limitation, and future work can optimize the interpolation step to realize these runtime gains. We will revise the claim in L121 to remove references to time efficiency and will include updated runtime results in the paper.
> - Crucially, on the other hand, from a memory footprint perspective—particularly in large-scale settings, as the reviewer noted—Q3C shows significant advantages over TD3. We conducted a detailed analysis comparing the memory usage of Q3C and TD3 across increasing network sizes. In this setup, we ensured that the number of critic parameters in Q3C matched those in TD3, allowing us to isolate the impact of the additional actor in TD3. As shown below, we observe increasing memory savings for Q3C as network size scales:
> | Model scale* | Q3C params | TD3 params | **Q3C memory (MB)** | **TD3 memory (MB)** | % Memory saved by removing actor |
> |--------------|---------------|----------------|---------------------|---------------------|------------------|
> | **Small**    | 37M         | 52M           | 341                 | 400                 | **14.8 %** |
> | **Medium**   | 135M          | 226M          | 707                 | 1,057               | **33.1 %** |
> | **Large**    | 538M         | 906M          | 2,245               | 3,675               | **38.9 %** |
> | **X-Large**  | 1.57B         | 2.65B          | 6,212               | 10,348              | **40.0 %** |
>
> \*Mapping of Layer Sizes: **Small → 1 024 control points, Medium → 2 048, Large → 4 096, X-Large → 7000**.
> Note: The critic used in Q3C and TD3 are normalized.
>
> Additionally, we conducted a proof-of-concept experiment where we match the total number of parameters in Q3C to those of TD3’s critic, keeping TD3’s architecture unchanged. Despite the parameter reduction, Q3C maintained comparable performance.
> | Method | Hopper-v4 |
> |-------|----------|
> | Q3C | 3050.92 ± 751.89 |
> | TD3 | 3113.41 ± 888.17 |
> We thank the reviewer for this insightful experiment, which helps us demonstrate the key benefit of Q3C in large-scale settings.

---

> > ### Comment · Reviewer_mqYt · 2025-08-06
> >
> > Thank you for conducting the additional experiments demonstrating the memory savings of Q3C compared to TD3. These results provide valuable insights and address the key concerns I had. The demonstrated efficiency could be a compelling reason to adopt this algorithm in large-scale settings. Please ensure that the paper is revised accordingly based on the rebuttal and our discussion. Since most of my comments were addressed satisfactorily, I have decided to raise my score slightly.

---

### Official Review · Reviewer_SR2K · 2025-07-02

**Clarity:** 2
**Significance:** 2
**Originality:** 2
**Rating:** 3
**Confidence:** 3

**Summary:**

The authors in this submission proposed a value-based framework for reinforcement learning with continuous actions. In contrast to the actor-critic methods, they first constructed the control points and subsequently learned policies with respect to Q-functions over those points. In order to overcome the issues reported in previous work, they proposed a few techniques to address them. The experiments showed that the proposed method is on par with TD3 in standard domains, and achieves better performance on constricted environments.

**Questions:**

* Could you comment on the complexity for Q3C, when compared against relevant value or policy based methods, given the additional steps w.r.t. control points?
* Why is the variance for Q3C in Table 2 on InvertedPendulum 0?

**Ethical Concerns:**

["NO or VERY MINOR ethics concerns only"]

**Final Justification:**

I've read the authors' response and some of my comments and questions have been addressed. I increased my score slightly. The memory improvement looks valid, while I am not confident about the time saving, which may depend on the exact implementation.

**Limitations:**

Yes

**Quality:**

3

**Strengths And Weaknesses:**

Strength:
The revisit of the Q-function approximation using the control-points is an interesting direction. The authors also showed that the proposed method, when combined with neural networks, works reasonablly well in some standard domains

Weakness:
Given the additional complexity involving the control points, the performance of the proposed Q3C algorithm is mostly similar to that of TD3. Note that TD3 is far behind some other policy gradient methods, e.g., SAC. Subsequently, its significance is a bit limited.

Some steps in the proposed method need justification and the presentation can be improved. A few of them are listed as follows.
* In L175-L179, the authors proposed to keep top k from N candidates, so why not just generate k control-points?
* In Eq. (6), should there be $N (N - 1) / 2$ pairs only?

---

> ### Author Rebuttal · Authors · 2025-07-31
>
> We thank the reviewer for their comments and suggestions. We’re glad to hear they found our method interesting and appreciated the broad range of environments we evaluated. Below, we address the concerns about the introduction of control points in Q3C, significance in comparison to TD3 and SAC, and justifications for algorithm design.
>
> ----
>
> ## Complexity: Control point architecture reduces, not adds to, learning complexity
>
> ### 1. Control points enable training a single end-to-end network instead of a separate actor and critic.
> Q3C reduces the learning complexity to simply learning a single Q-network by eliminating the actor network and its training objective completely. This single Q-network can be trained end-to-end because the control points function as just another layer of the neural network.
> While Q3C has two modules, for proposing and evaluating the control points, a **crucial difference from actor-critic** approaches is that these modules are trained end-to-end with a single Bellman objective. Thus, it removes the complexities seen in prior works such as DDPG, TD3, and SAC, of specifying the gradient objective for training the actor and hyperparameter tuning of entropy, policy delay, or dual optimizers. Fundamentally, the only architectural addition in Q3C is the parameter-free inverse-distance interpolator.
> ### 2. Q3C has a lightweight wall-time and memory footprint
> Contrary to concerns about additional complexity, Q3C simplifies training by eliminating the need for a separate actor network and its corresponding backward pass. We report wall-clock and PyTorch memory used versus baselines (5 seeds each) to show Q3C remains lightweight in practice.
>
> **Wall time until convergence**
>
> | Algorithm | Hopper | BipedalWalker | HalfCheetah | InvertedPendulumBox | HalfCheetahBox | HopperBox |
> |-----------|--------|---------------|-------------|---------------------|----------------|-----------|
> | Q3C | 58.3 min | **44.6 min** | 61.5 min | **12.43 min** | 4 hr | **0.7 hr** |
> | TD3 | **32.9 min** | 45.6 min | **39.2 min** | N/A | N/A | N/A |
> | SAC | 76.3 min | 50.5 min | 69.2 min | 16.5 min | 6.6 hr | 1.3 hr |
> | PPO | N/A | N/A | N/A | N/A | **1.0 hr** | N/A |
> | NAF | N/A | N/A | N/A | N/A | 7.2 hr | N/A |
>
>
> **Pytorch memory**
>
> | Algorithm | Pytorch Allocated (MB) |
> |-----------|------------------------|
> | Q3C       | 26.58                  |
> | TD3       | 23.51                  |
> | SAC       | 20.01                  |
> | PPO       | 18.38                  |
>
> ----
>
> ## Significance compared to TD3 and SAC
>
> ### 1. Note: TD3 is NOT far behind SAC
> We politely point out that TD3 is actually proven to be better than SAC in certain environments, as observed in prior work [1, 4, 5, 7], and there is **no general consensus** that SAC is the superior algorithm against TD3 [1, 2, 3, 8]. These two algorithms employ different exploration mechanisms and thus outperform each other often because of which style of exploration is more efficient in a given environment and which hyperparameters were tuned better [6, 9].
>
> ### 2. Clarify: Q3C is primarily compared to TD3 because it uses TD3’s fairly comparable noise-added exploration.
> Since the exploration mechanism is known to conflate the outcomes of an algorithm in different environments [10, 11], all the baselines in the paper: TD3, NAF, Wire-fitting, and Q3C were run with the same noise-added exploration scheme. This helped us delineate the effect of the learning algorithm fairly.
> Nevertheless, as suggested by the reviewer, we have added the results on SAC and PPO, which are not directly comparable due to a significantly different experiment backbone, i.e., PPO is on-policy and SAC has learned exploration.
>
> ### 3. Added experiments on SAC, PPO, RBF-DQN
> As suggested by the reviewer, we have added SAC and PPO, as well as RBF-DQN recommended by Reviewer UQVA, as baselines. These methods were trained for the same number of timesteps as Q3C and TD3, which we also include for reference. SAC and PPO were implemented using Stable-Baselines3 (SB3) and trained with hyperparameters from RL Zoo3. For RBF-DQN, we used the official implementation and hyperparameters provided by the authors.
>
> | Environment | Q3C | TD3 | SAC | PPO | RBF-DQN |
> |-------------|-----|-----|-----|-----|---------|
> | Pendulum-v1 | -159.53 ± 16.46 | -144.64 ± 95.28 | -140.31 ± 31.14 | -176.85 ± 23.70 | 159.68 ± 15.80 |
> | Hopper-v4 | 3206.14 ± 407.23 | 3113.41 ± 888.17 | 2922.56 ± 590.18 | 882.85 ± 286.17 | 1912.34 ± 1203.35 |
> | Walker2d-v4 | 3977.39 ± 879.70 | 4770.82 ± 560.16 | 4091.98 ± 405.00 | 2917.58 ± 895.72 | 903.62 ± 308.31 |
> | HalfCheetah-v4 | 9468.66 ± 949.01 | 9984.74 ± 1076.58 | 10909.02 ± 416.07 | 5397.83 ± 746.90 | 7267.01 ± 1546.27 |
> | InvertedPendulumBox-v4 | 1000 ± 0 | 782.76 ± 348.92 | 993.40 ± 7.15 | 839.41 ± 372.97 | 734.83 ± 560.40 |
> | HalfCheetahBox-v4 | 4357.82 ± 1503.33 | 2276.70 ± 2036.59 | 4248.83 ± 2184.33 | 4637.60 ± 686.19 | 3023.26 ± 1980.03 |
> | HopperBox-v4 | 1974.28 ± 1170.05 | 1406.83 ± 1162.72 | 3330.64 ± 140.25 | 637.12 ± 449.74 | 1225.27 ± 793.74 |
>
> While SAC and PPO perform well in certain environments, we observe that TD3 outperforms both in others—a finding consistent with prior benchmarking studies [1, 4, 5, 7]. Q3C achieves competitive performance in all standard environments and matches or surpasses the state-of-the-art in all restricted environments. In contrast, the newly added baselines exhibit varying performance across different environments. This supports the claim that Q3C is a robust and effective alternative to actor-critic methods.
>
> ----
>
> ## Questions on top-k, Eq. 6, results variance.
>
> ### 1. Ablation experiment to validate top-k: Fig. 6, Appendix D.4
>
> We refer the reviewer to the results on Q3C - Ranking ablation in Figure 6 of the main paper and a general control point ablation in Appendix D.4. Here, we show that keeping top-k from N control-points outperforms generating k control points directly (this is equivalent to setting n=k).
>
> Discussion added to paper revision: This is because the number of control points n and the top-k value k serve key, different purposes when constructing the Q-function. Increasing the value of n gives us more coverage across high-dim action spaces and provides more representational potential as a larger number of control points can generally model more complex and vast Q-landscapes. The hyperparameter k is not tuned for overarching expressivity of the function, but for enforcing locality and allowing for piecewise approximation of the Q-function. By utilizing only the k closest points to the state-action pair we are approximating, we ensure that only control points that are relevant to that subset of action space contribute to the estimated Q-value, improving accuracy. This also ensures that control point values that lie very far in the action space are not distorted by training experiences on a very different action subset. While some of this distance weighting is already performed by our interpolator, selecting the top k control points enforces this more firmly and yields better results in terms of both performance and computational cost.
>
> ### 2. Equation 6 Clarification.
>
> For the separation loss used in Equation (6), minimizing the sum of pairwise inverse distances encourages the control points to spread out across the action space. In fact, this formulation effectively minimizes the maximum pairwise overlap, and the optimal solution corresponds to a uniform distribution of control points.
>
> ### 3. Variance in InvertedPendulumBox.
>
> The zero variance was incidental: Table 2 is consistent with the converged performances in Figure 4. Thus, since Q3C consistently achieved a score of 1000 across all 10 random seeds, the variance of the result is zero.
>
> ----
>
> **References**
>
> [1] Huang, Shengyi, et al. "Open RL Benchmark: Comprehensive Tracked Experiments for Reinforcement Learning." arXiv preprint arXiv:2402.03046 (2024).
> [2] Liu, Shijie. "An Evaluation of DDPG, TD3, SAC, and PPO: Deep Reinforcement Learning Algorithms for Controlling Continuous Systems." DAI 2023 (2024).
> [3] Mock, James W., and Suresh S. Muknahallipatna. "A Comparison of PPO, TD3 and SAC Reinforcement Algorithms for Quadruped Walking Gait Generation." Journal of Intelligent Learning Systems and Applications (2023).
> [4] Jain, Ayush, et al. "Mitigating Suboptimality of Deterministic Policy Gradients in Complex Q-Functions." Reinforcement Learning Conference.
> [5] Moosa, Farhana, et al. "Benchmarking Reinforcement Learning Algorithms for Autonomous Mechanical Thrombectomy." International Journal of Computer Assisted Radiology and Surgery (2025).
> [6] Wan, Yi, Dmytro Korenkevych, and Zheqing Zhu. "An Empirical Study of Deep Reinforcement Learning in Continuing Tasks." arXiv preprint arXiv:2501.06937 (2025).
> [7] Rzayev, Altun, and Vahid Tavakol Aghaei. "Off-Policy Deep Reinforcement Learning Algorithms for Handling Various Robotic Manipulator Tasks." arXiv preprint arXiv:2212.05572 (2022).
> [8] Achiam, Joshua. "Spinning Up in Deep Reinforcement Learning." OpenAI (2018).
> [9] Haarnoja, Tuomas, et al. "Soft Actor-Critic: Off-Policy Maximum Entropy Deep Reinforcement Learning with a Stochastic Actor." ICML (2018).
> [10] Balloch, Jonathan C., et al. "Is Exploration All You Need? Effective Exploration Characteristics for Transfer in Reinforcement Learning." arXiv preprint arXiv:2404.02235 (2024).
> [11] Su, Yi, et al. "Long-term value of exploration: measurements, findings and algorithms." Proceedings of the 17th ACM International Conference on Web Search and Data Mining. 2024.
>
> ----
>
> We hope the clarification on reduced learning complexity and new results adequately address your concerns by demonstrating the theoretical and practical significance of Q3C. We are happy to answer any further questions and incorporate any further suggestions.

---

> > ### Comment · Reviewer_SR2K · 2025-08-07
> >
> > Thank you for providing the rebuttal to my questions and comments. It's not entirely clear if the comparison is apple-to-apple, as running time for some experiments is lacking. Nevertheless, I increase my score to acknowledge the improvement compared with the earlier version.

---

> ### Comment · Area_Chair_aKoN · 2025-08-06
> **Please Read Authors' Rebuttal and Respond ASAP**
>
> There are only two days remaining in the author-reviewer discussion phase. The authors have made a concerted effort to address your concerns, and it's important that you respond to let them know whether their rebuttal addresses the issues that you raised.

---

> ### Author Response · Authors · 2025-08-07
> **Important Clarification: Missing running time just means those baselines were unsuccessful**
>
> We thank the reviewer for their response and for reconsidering their score. We want to make an important clarification that the runtime comparisons are not lacking — **N/A means that the algorithm did not meet the set performance threshold to be considered a successful run**.
>
> ## Clarify: N/A does not mean incomplete experiment
>
> To compare different algorithms apples-to-apples, we had defined a reward threshold for each environment (Hopper: 3000, Bipedal Walker: 300, HalfCheetah: 9000, InvertedPendulumBox: 1000, HalfCheetahBox: 4000, HopperBox: 2000) corresponding to what is generally considered optimal performance. We then recorded the time each algorithm took to reach and stabilize around that threshold. For algorithms marked as “N/A,” this indicates that they did not reach or maintain the threshold performance within the full training duration. We have relabeled "N/A" as "No success" to be precise.
>
> | Algorithm | Hopper | BipedalWalker | HalfCheetah | InvertedPendulumBox | HalfCheetahBox | HopperBox |
> |-----------|--------|---------------|-------------|---------------------|----------------|-----------|
> | **Q3C**   | 58.3 min | **44.6 min** | 61.5 min | **12.43 min** | 4 hr | **0.7 hr** |
> | **TD3**   | **32.9 min** | 45.6 min | **39.2 min** | No success† | No success† | No success† |
> | **SAC**   | 76.3 min | 50.5 min | 69.2 min | 16.5 min | 6.6 hr | 1.3 hr |
> | **PPO**   | No success† | No success† | No success† | No success† | **1.0 hr** | No success† |
> | **NAF**   | No success† | No success† | No success† | No success† | 7.2 hr | No success† |
>
> † Run completed but did not reach the required performance threshold.
>
> ## Time and Memory Savings of Q3C
>
> **Runtime**: From a runtime perspective, Q3C removes the actor network, which reduces computational overhead. However, this advantage is partially offset by the additional cost introduced by the interpolation mechanism. Since we’ve used a simple interpolation module, its overhead cancels out the time savings from removing the actor. In future work, we would optimize it with better parallelism and alternative implementations. That said, we believe this is an engineering limitation, and future work can optimize the interpolation step to realize these runtime gains. We will revise the claim in L121 to remove references to time efficiency and will include updated runtime results in the paper.
>
> **Memory footprint**: Crucially, on the other hand, from a memory footprint perspective—particularly in large-scale settings, as the reviewer noted—Q3C shows significant advantages over TD3. We conducted a detailed analysis comparing the memory usage of Q3C and TD3 across increasing network sizes. In this setup, we ensured that the number of critic parameters in Q3C matched those in TD3, allowing us to isolate the impact of the additional actor in TD3. As shown below, we observe increasing memory savings for Q3C as network size scales:
> | Model scale* | Q3C params | TD3 params | **Q3C memory (MB)** | **TD3 memory (MB)** | % Memory saved by removing actor |
> |--------------|---------------|----------------|---------------------|---------------------|------------------|
> | **Small**    | 37M         | 52M           | 341                 | 400                 | **14.8 %** |
> | **Medium**   | 135M          | 226M          | 707                 | 1,057               | **33.1 %** |
> | **Large**    | 538M         | 906M          | 2,245               | 3,675               | **38.9 %** |
> | **X-Large**  | 1.57B         | 2.65B          | 6,212               | 10,348              | **40.0 %** |
>
> \*Mapping of Layer Sizes: **Small → 1024 control points, Medium → 2048, Large → 4096, X-Large → 7000**.
> Note: The critic used in Q3C and TD3 are normalized.
>
> ----
>
> We thank the reviewer once again for their feedback, engagement, and important questions. We hope this response clarifies that Q3C has a lightweight wall-time and a favorable memory footprint, strengthening the understanding of Q3C’s advantages over the baselines. We hope this addresses the reviewer's remaining concerns.

---

### Official Review · Reviewer_UQVA · 2025-07-03

**Clarity:** 4
**Significance:** 2
**Originality:** 2
**Rating:** 5
**Confidence:** 5

**Summary:**

This paper proposes Q-learning for Continuous Control with Control-points (Q3C), a value-based method for control in domains with continuous action spaces. This approach structures the Q-function approximator such that it covers the entire continuous space, but the maximizing action can still be determined efficiently. The authors prove that this method retains the universal approximator property of the underlying neural network models and that it compares favorably to some other value-based methods in several continuous control domains.

**Questions:**

How would you present Q3C in a context including RBF-DQN?

What would you say are the signfiicant differences between Q3C and RBF-DQN and how do you think they improve on RBF-DQN?

Can you add RBF-DQN as a baseline?

Is there a principled (or inspired-by-principle) way of selecting the number of control points used for a problem?

**Ethical Concerns:**

["NO or VERY MINOR ethics concerns only"]

**Final Justification:**

The authors addressed my concerns related to the lack of comparison/discussion regarding RBF-DQN, which is a very similar approach which was published in 2021. However, with the authors adding this discussion and adding RBF-DQN as a baseline to their experimental results, I believe that the paper provides novel, useful additions to such approaches.

**Limitations:**

I would like to see more discussion about how to select the number of control points to use. While the theory contribution shows that there exists an N such that any arbitrary maximum error can be guaranteed, such an N could be quite large for domains with complex value functions.

**Quality:**

3

**Strengths And Weaknesses:**

# Strengths

The paper is well-written and easy to follow.

The problem of value-based continuous control is an important one that I believe is relevant and of interest to the community.

Q3C's control point diversity and control point ranking components appear novel and useful additions to the algorithm.

Q3C performs well compared to the baselines the authors use and the experiments thoroughly cover a large number of interesting domains, though key baselines are missing (see Weaknesses).


# Weaknesses

Q3C is very similar to RBF-DQN [1], but it the latter is neither mentioned in either the text nor used as a baseline for the experiments. RBF-DQN uses the same kind of structure to generate candidate actions (centroids there) and values as functions of the state. These are then interpolated, but using radial basis functions explicitly. Equation 4 for Q3C is essentially the same as Equation 5 in the RBF-DQN paper, but uses something like an inverse quadratic RBF where RBF-DQN uses negative exponential RBF. RBF-DQN similarly retains the universal approximator property and the approximator is maximized (with bounded error) by selecting the maximizing centroid.

Q3C is not compared to policy-based methods (policy gradient or actor-critic) in the experiments.

# References
[1]  Deep Radial-Basis Value Functions for Continuous Control, Asadi et al., AAAI 2021

---

> ### Author Rebuttal · Authors · 2025-07-31
>
> We thank the reviewer for recognizing the novelty of our additions to the control-point-based Q-function approximator and for appreciating the wide variety of our evaluation environments.
>
> ----
>
> ## Added comparison to RBF-DQN
>
> We appreciate the pointer to RBF-DQN (Asadi et al., AAAI’21) and agree it should be appropriately cited, discussed, and compared. Thus, we (i) implement and evaluate RBF-DQN across all environments using its official code with the same exploration mechanism used for all our reported results, and (ii) include a discussion to situate Q3C among continuous-control, value-based interpolation methods.
>
> **(i) Q3C empirically outperforms RBF-DQN**
> | Environment              | Q3C                     | RBF-DQN                |
> |--------------------------|--------------------------|-------------------------|
> | Pendulum-v1              | -159.53 ± 16.46          | -159.68 ± 15.80          |
> | Hopper-v4                | 3206.14 ± 407.23         | 1912.34 ± 1203.35       |
> | Walker2d-v4              | 3977.39 ± 879.70         | 903.62 ± 308.31         |
> | HalfCheetah-v4           | 9468.66 ± 949.01         | 7267.01 ± 1546.27       |
> | InvertedPendulumBox-v4   | 1000 ± 0                 | 734.83 ± 560.40         |
> | HalfCheetahBox-v4        | 4357.82 ± 1503.33        | 3023.26 ± 1980.03       |
> | HopperBox-v4             | 1974.28 ± 1170.05        | 1225.27 ± 793.74        |
>
> **(ii) Added discussion in paper to contextualize Q3C among value-based interpolation methods like RBF-DQN**
>
> ### Similarities between Q3C and RBF-DQN
> - Both approaches address the problem of value-based RL for a continuous action space without an actor.
> - Both generate a finite set of action anchor points, which help construct an interpolation over Q-values.
> - The maximizing action is chosen easily from among the action anchor points.
>
> To assign appropriate credit to RBF-DQN, we have revised the manuscript to include it alongside the body of work on wire-fitting interpolators, which is the key inspiration and baseline of Q3C. A key benefit of RBF-DQN over Q3C could be its stronger approximation ability due to the use of explicit radial basis functions. However, since reinforcement learning only requires the Q-function to be maximal for the optimal action, the benefit of a more accurate Q-function may not necessarily result in sample-efficient learning.
>
> We clearly contextualize the structural maximization in Q3C and contrast it to approximate maximization (under bounded error) of RBF-DQN, along with other key differences below.
>
> ### Differences between Q3C and RBF-DQN
>
> While both Q3C and RBF-DQN use interpolation techniques, there are key differences in design and practical implications:
> Interpolation Scheme: RBF-DQN employs radial basis functions with negative exponential kernels. In their standard form, these do not guarantee that the interpolated function attains its maximum at one of the basis centroids. To mitigate this, RBF-DQN introduces an inverse smoothing parameter to push the maximum toward a centroid, albeit with bounded approximation error. However, this smoothing also spreads high values to nearby actions, smoothing the curve and making it less suitable in scenarios with multiple local optima.
>
> In contrast, Q3C uses inverse-distance weighting, which guarantees that the maximum Q-value occurs at one of the control points. The smoothing parameter in our formulation affects the function's value only at other control points: with a low smoothing parameter, the approximated function passes through all control points, while with a large smoothing parameter, the function's value may differ slightly at some control points. However, in all cases, it is still guaranteed that the maximum of the function lies at a control point—an advantage in highly non-convex Q landscapes.
>
> These theoretical differences are also reflected in our empirical findings:
> - Q3C consistently outperforms RBF-DQN in standard environments and shows even stronger gains in restricted settings, where the Q-function is discontinuous or non-smooth. RBF-DQN struggles in these cases, which supports our claim that smoothing-based interpolation may be a poor fit for such landscapes.
> - Additionally, **Q3C is more parameter-efficient**. Most of our experiments use only 20 control points, whereas RBF-DQN uses at least 100 centroids.
>
> Beyond the core interpolation strategy, Q3C introduces several novel architectural and algorithmic enhancements, including:
> - Action-conditioned Q-value generation, which improves representational capacity;
> - Relevance-based top-K control point filtering;
> - Q-value normalization for scale-awareness;
> - A light entropy-inspired regularization term for training stability.
>
> These enhancements are validated through ablations in the paper, and several of them could potentially benefit RBF-DQN as well, which adds to the significance of our paper.
>
> ----
>
> ## Added policy-based comparisons: SAC, PPO
> We also included SAC and PPO as additional baselines, and re-report TD3 for comparison. The results are summarized below:
> | Environment | Q3C | TD3 | SAC | PPO | RBF-DQN |
> |-------------|-----|-----|-----|-----|---------|
> | Pendulum-v1 | -159.53 ± 16.46 | -144.64 ± 95.28 | -140.31 ± 31.14 | -176.85 ± 23.70 | 159.68 ± 15.80 |
> | Hopper-v4 | 3206.14 ± 407.23 | 3113.41 ± 888.17 | 2922.56 ± 590.18 | 882.85 ± 286.17 | 1912.34 ± 1203.35 |
> | Walker2d-v4 | 3977.39 ± 879.70 | 4770.82 ± 560.16 | 4091.98 ± 405.00 | 2917.58 ± 895.72 | 903.62 ± 308.31 |
> | HalfCheetah-v4 | 9468.66 ± 949.01 | 9984.74 ± 1076.58 | 10909.02 ± 416.07 | 5397.83 ± 746.90 | 7267.01 ± 1546.27 |
> | InvertedPendulumBox-v4 | 1000 ± 0 | 782.76 ± 348.92 | 993.40 ± 7.15 | 839.41 ± 372.97 | 734.83 ± 560.40 |
> | HalfCheetahBox-v4 | 4357.82 ± 1503.33 | 2276.70 ± 2036.59 | 4248.83 ± 2184.33 | 4637.60 ± 686.19 | 3023.26 ± 1980.03 |
> | HopperBox-v4 | 1974.28 ± 1170.05 | 1406.83 ± 1162.72 | 3330.64 ± 140.25 | 637.12 ± 449.74 | 1225.27 ± 793.74 |
>
> PPO is on-policy, so it is usually sample-inefficient. SAC employs a different exploration scheme based on soft Q-value maximization in contrast to the noise-added exploration used in the baselines in the paper: TD3, NAF, wire-fitting, Q3C, and now RBF-DQN. Thus, these methods are not directly comparable to evaluate the key contribution of Q3C as a “structurally maximizable Q-function”, and we do NOT expect or claim that Q3C should outperform these methods.
>
> ----
>
> ## Selecting number of control points
>
> Regarding control point count: while it is a hyperparameter, our ablation studies (Appendix D.4) show that simply increasing the number of control points n does not always improve performance. A too large n runs the risk of overfitting or unnecessarily increasing the problem complexity which slows down convergence. A typical recipe is to start with the default n=20 and try larger numbers of control points with a fixed top-k filter (k=10), which balances coverage and computational cost, leading to robust and stable performance.
>
> We find that the optimal number of control points tends to scale more conservatively than a proportional increase with action dimensionality due to the constant tradeoff between increasing representational potential and increasing the learning difficulty for the model. However, a limited number of additional control points are beneficial for environments with increased general task complexity and a considerable higher action dimensionality. With relatively complicated environments such as Ant, HalfCheetahBox, and Adroit, we found an optimal control point combination to be between n=30,40,50 and k=10.
>
> For environments with an even higher dimensionality, we can utilize a larger n with relatively minimal increases to computational cost and problem complexity as the backpropagation still only needs to optimize across the top k control points in a single backward pass. A key benefit of the conditional Q architecture is that even with a large number of control points, the number of parameters or the compute required does not increase because it is applied parallel over all the control points. This is also a major representational benefit over RBF-DQN.
>
> ---
>
> We thank the reviewer once again for their valuable comments, which helped us improve our work. We will incorporate these details and the supporting experimental results into the final version of our paper. As the comparison with RBF-DQN was the reviewer’s primary concern, we have now included this baseline along with a detailed discussion and new experimental results. Given these additions and clarifications, we kindly ask the reviewer to consider revisiting their score, and we are happy to answer any further questions.

---

> > ### Comment · Reviewer_UQVA · 2025-08-07
> >
> > Thank you for your responses to my concerns. My main concern was the lack of discussion of RBF-DQN given the similarity between it and Q3C, and you have addressed that by comparing and contrasting the two approaches and adding RBF-DQN as a baseline. I expect those will be added to the final version of the paper.
> >
> > I have also looked over the responses to the other reviewers and have gained useful insight from those as well.
> >
> > I have raised my rating accordingly.

---

> ### Comment · Area_Chair_aKoN · 2025-08-06
> **Please Read Authors' Rebuttal and Respond ASAP**
>
> There are only two days remaining in the author-reviewer discussion phase. The authors have made a concerted effort to address your concerns, and it's important that you respond to let them know whether their rebuttal addresses the issues that you raised.

---

### Official Review · Reviewer_U2v4 · 2025-07-05

**Clarity:** 4
**Significance:** 2
**Originality:** 3
**Rating:** 5
**Confidence:** 4

**Summary:**

This paper introduces Q3C, a novel actor-free Q-learning approach for continuous control, aiming to overcome the stability issues associated with actor-critic methods. Traditional actor-critic methods suffer from instability during training due to the coupling of policy and value function updates. The authors propose a solution using a structurally maximizable Q-function, optimized by learning control points. By leveraging deep learning and a set of architectural and algorithmic improvements, Q3C is designed to be stable and sample-efficient.

**Questions:**

1. The paper suggests that actor-critic methods are generally stable, but the proposed method also involves training two models: one for the policy and another for the value function. This dual-model structure could introduce potential instability, as the training of both models might become misaligned. Could the authors elaborate on how stability is maintained in their method despite the dual-model setup? What specific mechanisms or architectural choices help ensure training stability, and how does Q3C compare with traditional actor-critic methods in this regard?

2. The paper compares Q3C primarily with value-based methods. To gain a better understanding of its performance and robustness, would the authors consider adding comparisons to stronger policy-gradient methods like PPO? How do the proposed method’s strengths and weaknesses compare with these baselines, particularly in terms of exploration, sample efficiency, and policy robustness?
Generalization to High-Dimensional Action Spaces:
How does Q3C scale to environments with high-dimensional action spaces? Are there any performance bottlenecks in optimizing the Q-function in such spaces? Additionally, how well does the method handle partial observability or environments where the action space is not fully observable?

3. Given that Q3C does not use an explicit actor for exploration, how does the method ensure sufficient exploration during training? Would additional exploration mechanisms, such as Boltzmann over the control-point values, improve performance in environments that require more extensive exploration?
Real-World Applicability and Sample Efficiency:
While the paper provides strong empirical results in simulation environments, has there been any consideration of the method's applicability in real-world settings, especially for high-dimensional or physical tasks? Additionally, Q3C’s sample efficiency could be compared in greater detail with methods such as SAC or TD3, especially in tasks with sparse rewards or large state spaces.

**Ethical Concerns:**

["NO or VERY MINOR ethics concerns only"]

**Final Justification:**

The proposed method is technically robust, supported by clear theoretical motivation and solid empirical validation. The integration of control points for Q-function maximization is a novel and effective approach, enabling efficient identification of the maximum in continuous action spaces. The authors also clearly articulate the scope and contributions of the paper.

**Limitations:**

Yes. The paper acknowledges that Q3C’s performance may suffer in certain high-dimensional or sparse-reward environments, and it notes that further exploration of the method’s sample efficiency and real-world scalability would be useful. However, a more detailed discussion of the computational limitations or potential scalability challenges in real-world applications would provide a more comprehensive understanding.

**Paper Formatting Concerns:**

Nan

**Quality:**

3

**Strengths And Weaknesses:**

**Strengths:**

1. The method proposed is technically robust, with clear theoretical motivation and sound empirical validation.
The integration of control points for Q-function maximization is a novel approach, ensuring that the maximum value can be found efficiently in continuous action spaces, a known challenge in Q-learning.

2. The variance reduction techniques, such as exponential moving averages (EMA), effectively address training instability, ensuring better convergence.

3. Empirical results show Q3C is competitive with leading actor-critic methods such as TD3, SAC, and others, performing well across multiple environments, including complex and constrained settings.

**Weaknesses:**

1. While the concept of using control points is not entirely new, the combination of this approach with deep learning and the specific modifications introduced by the authors is what makes this work original.
However, it would be useful to see a deeper comparison of how Q3C outperforms or offers advantages over other actor-free methods, such as Implicit Q-learning or Policy-conditioned Q-learning, which also aim to address similar problems in continuous control.

2. The paper suggests that actor-critic methods are generally stable, but the proposed method also involves training two models: one for the policy and another for the value function. This dual-model structure could introduce potential instability, as the training of both models might become misaligned. Could the authors elaborate on how stability is maintained in their method despite the dual-model setup? What specific mechanisms or architectural choices help ensure training stability, and how does Q3C compare with traditional actor-critic methods in this regard?

3. The paper compares Q3C primarily with value-based methods. To gain a better understanding of its performance and robustness, would the authors consider adding comparisons to stronger policy-gradient methods like PPO? How do the proposed method’s strengths and weaknesses compare with these baselines, particularly in terms of exploration, sample efficiency, and policy robustness?
Generalization to High-Dimensional Action Spaces:
How does Q3C scale to environments with high-dimensional action spaces? Are there any performance bottlenecks in optimizing the Q-function in such spaces? Additionally, how well does the method handle partial observability or environments where the action space is not fully observable?

4. Given that Q3C does not use an explicit actor for exploration, how does the method ensure sufficient exploration during training? Would additional exploration mechanisms, such as Boltzmann over the control-point values, improve performance in environments that require more extensive exploration?
Real-World Applicability and Sample Efficiency:
While the paper provides strong empirical results in simulation environments, has there been any consideration of the method's applicability in real-world settings, especially for high-dimensional or physical tasks? Additionally, Q3C’s sample efficiency could be compared in greater detail with methods such as SAC or TD3, especially in tasks with sparse rewards or large state spaces.

---

> ### Author Rebuttal · Authors · 2025-07-31
>
> We appreciate the reviewer’s thoughtful comments and positive assessment of our method’s originality, as well as the value of our theoretical and empirical validations. Below, we address the all the concerns and questions, with added experiments wherever possible.
>
> ## Comparison of Q3C with IQL
>
> Our work focuses on employing control-point-based Q-function interpolation in the context of online reinforcement learning, and we compare against other online RL methods accordingly. While Implicit Q-Learning (IQL) is a strong offline RL algorithm, it is not directly applicable to online settings and thus not included as a baseline. However, future work could explore the utility of our Q-function architecture in offline RL settings.
>
> Regarding policy-conditioned Q-learning, we would appreciate clarification or a reference from the reviewer, as we have not been able to find a method by this name in the online RL literature.
>
> ## Difference between Q3C’s framework and Actor-Critic Methods
> Q3C’s control points enable training a single end-to-end network instead of a separate actor and critic, because the control points function as just another layer of the neural network. While Q3C has two modules, for proposing and evaluating the control points, a **crucial difference from actor-critic** approaches is that these modules are trained end-to-end with a single Bellman objective.
>
> Although this setup resembles actor-critic methods at first, it differs significantly: Q3C uses a single, unified loss function to train both components end-to-end, without the alternating optimization of separate actor and critic models. Thus, it retains a purely value-based formulation.
>
> Thus, it removes the complexities seen in prior works such as DDPG, TD3, and SAC, of specifying the gradient objective for training the actor and hyperparameter tuning of entropy, policy delay, or dual optimizers. Fundamentally, the only architectural addition in Q3C is the parameter-free inverse-distance interpolator. We have revised the manuscript to clearly establish this distinction.
> ## Comparison with policy-based methods: SAC, PPO
> As the reviewer recommended, we have added SAC and PPO, as well as RBF-DQN suggested by Reviewer UQVA, as new baselines. These were trained for an equal number of timesteps as Q3C and TD3. For SAC and PPO, we use the Stable-Baselines3 (SB3) implementation and hyperparameters from RL Zoo3. For RBF-DQN, we rely on the official implementation and recommended hyperparameters provided by the authors. We report the final performance over 10 trials.
>
> | Environment | Q3C | TD3 | SAC | PPO | RBF-DQN |
> |-------------|-----|-----|-----|-----|---------|
> | Pendulum-v1 | -159.53 ± 16.46 | -144.64 ± 95.28 | -140.31 ± 31.14 | -176.85 ± 23.70 | 159.68 ± 15.80 |
> | Hopper-v4 | 3206.14 ± 407.23 | 3113.41 ± 888.17 | 2922.56 ± 590.18 | 882.85 ± 286.17 | 1912.34 ± 1203.35 |
> | Walker2d-v4 | 3977.39 ± 879.70 | 4770.82 ± 560.16 | 4091.98 ± 405.00 | 2917.58 ± 895.72 | 903.62 ± 308.31 |
> | HalfCheetah-v4 | 9468.66 ± 949.01 | 9984.74 ± 1076.58 | 10909.02 ± 416.07 | 5397.83 ± 746.90 | 7267.01 ± 1546.27 |
> | InvertedPendulumBox-v4 | 1000 ± 0 | 782.76 ± 348.92 | 993.40 ± 7.15 | 839.41 ± 372.97 | 734.83 ± 560.40 |
> | HalfCheetahBox-v4 | 4357.82 ± 1503.33 | 2276.70 ± 2036.59 | 4248.83 ± 2184.33 | 4637.60 ± 686.19 | 3023.26 ± 1980.03 |
> | HopperBox-v4 | 1974.28 ± 1170.05 | 1406.83 ± 1162.72 | 3330.64 ± 140.25 | 637.12 ± 449.74 | 1225.27 ± 793.74 |
>
> As shown, Q3C achieves competitive performance in all standard environments and matches or surpasses the state-of-the-art in all restricted environments. In contrast, the newly added baselines exhibit varying performance across different environments.
> ## High-dimensional Action Spaces and Partial Observability
> We have conducted preliminary experiments on two environments with high-dimensional action spaces. Without any hyperparameter optimization, runs of Q3C are able to match the final performance of TD3 in AdroitHandHammer (26-dim action space) and AdroitHandDoor (28-dim action space) environments, which shows that Q3C is able to scale its learning to high-dimensional actions.
> **Final Eval Success Rate on Adroit Envs**
>
> | Environment | Q3C | TD3 |
> |-------------|-----|-----|
> | AdroitHandHammer-v1 | 0.9 | 0.9 |
> | AdroitHandDoor-v1 | 0.92 | 0.84 |
>
>
> **Regarding the reviewer’s question about non-observable action spaces:** if this refers to partial observability in the state space, our algorithm remains unaffected, as our method does not impose any assumptions requiring full observability. If the concern is about action spaces with unavailable or invalid actions, our restricted environments explicitly model such scenarios, and we demonstrate that Q3C performs robustly under these conditions (Section 5.2). If the reviewer meant something else, we would appreciate further clarification.
>
> ## Exploration strategy of Q3C
>
> For exploration, we refer the reviewer to Algorithm 1 in the main paper. We use Gaussian noise added to the maximizing action, consistent with TD3, to isolate the effect of the learning algorithm from the exploration strategy. This ensures a fair comparison. We also experimented with Boltzmann exploration over the control points, but found that Gaussian noise resulted in more stable and consistent performance. We will add those comparisons to the final version of our paper.
>
> ## Sample Efficiency of Q3C
>
> In terms of sample efficiency, Q3C performs comparably to other baselines. As a hybrid between DQN-style and actor-critic methods, it offers the flexibility to incorporate improvements from both paradigms. In Appendix 5.3, we explored one such strategy of Cross-Q [1] and observed promising preliminary results. Future work can build on this direction to further improve sample efficiency. Additionally, our initial experiments on Adroit dexterous manipulation tasks indicate that Q3C can handle high-dimensional action spaces, suggesting potential for future applications in real-world scenarios that combine both scalability and efficiency.
>
> **References**
>
> [1] Bhatt, Aditya, et al. "Crossq: Batch normalization in deep reinforcement learning for greater sample efficiency and simplicity." arXiv preprint arXiv:1902.05605 (2019).
>
> ---
> We thank you for your feedback and hope we've addressed your comments adequately. We are happy to answer any further questions and incorporate any further suggestions.

---

> > ### Comment · Reviewer_U2v4 · 2025-08-07
> >
> > Thank you for the detailed response. I’ve decided to raise my score.

---

> ### Comment · Area_Chair_aKoN · 2025-08-06
> **Please Read Authors' Rebuttal and Respond ASAP**
>
> There are only two days remaining in the author-reviewer discussion phase. The authors have made a concerted effort to address your concerns, and it's important that you respond to let them know whether their rebuttal addresses the issues that you raised.

---

### Author Response · Authors · 2025-08-05

We would like to kindly request the reviewers to check our posted detailed rebuttals and updated experimental results. We sincerely appreciate the thoughtful feedback provided so far and would be grateful if the reviewer could take a moment to review our responses. Please feel free to let us know if there are any unaddressed concerns — we would be happy to elaborate.

---

### Note · Authors · 2025-08-12

We thank all the reviewers and AC for their constructive engagement and time. The discussions during the rebuttal helped improve the paper and clarify all the reviewers’ concerns, as they acknowledged in their final responses. Here, we summarize the revisions made to our work:

- **Baselines**: Added requested comparisons to
    - actor-critic approaches — **SAC, PPO** — Q3C **reduces complexity** compared to actor-critic by training a single end-to-end model [`U2v4`, `SR2K`, `mqYt`],
    - actor-free approaches — **RBF-DQN** — Q3C is more **parameter-efficient** and RBF-DQN is susceptible to smoothing error [`UQVA`].
- **Scalability**:
    - Q3C scales to **high-dimensional actions**, like 26-dim Adroit Hammer [`U2v4`]
    - Q3C has **increasing memory savings** over TD3 as the model size grows [`mqYt`, `SR2K`]
    - Q3C has a **lightweight wall-clock time** [`SR2K`, `mqYt`]

----

**Clarifying `SR2K` remaining comment**: Crucially, we would like to highlight that the last remaining concern of `SR2K` is a misunderstanding that the runtime comparison table is *incomplete* due to N/A entries. We clarify that, in fact, the table is fully populated — **N/A means ''no success''**, i.e., a baseline did not reach the set reward threshold to be considered a successful run in that environment. This is an **even stronger result for Q3C** as these baselines are unable to reach Q3C performance.

---

### Decision · Program_Chairs · 2025-09-17

**Decision:**

Accept (poster)

**Comment:**

The paper proposes Q3C, an actor-free Q-learning framework for reinforcement learning (RL) in continuous action space settings. Unlike traditional actor-critic methods, which can be prone to training instability due to the tight coupling of the actor and critic updates, Q3C relies purely on value-based learning by structurally maximizing Q-functions. The core idea behind Q3C is to approximate the Q-function using a set of learned control points, implemented via a wire-fitting interpolator. This allows the problem of action maximization to be reduced to selecting among these candidates, which then avoids the need for a separately trained actor network. The paper proposes several architectural/algorithmic modifications suited to deep RL settings that improve the stability and efficiency of training. Empirical results across a range of continuous control benchmarks show that Q3C achieves performance and sample efficiency comparable to state-of-the-art actor-critic methods such as TD3, while significantly outperforming them in constrained action-space environments where traditional gradient-based maximization struggles.

The paper was reviewed by four referees who are in agreement regarding the key strengths and weaknesses of the paper as originally submitted. Among the strengths, the reviewers agree that the paper addresses a long-standing problem in Q-learning---Q-function maximization---and the means by which it does so through the integration of control points is both novel and interesting. Meanwhile, at least two reviewers note their appreciation for the theoretical motivation behind the proposed Q3C framework. While the reviewers acknowledge that the experimental evaluation provided in the submission demonstrates that Q3C is competitive with leading actor-critic baselines, several reviewers found that the paper would benefit from comparisons to other methods including PPO, SAC, and the related RBF-DQN method, which was neither cited nor compared against. Additionally, some reviewers raised questions about the computational cost associated with Q3C, suspecting that the use of control points introduces added complexity. The authors address these concerns in their rebuttal by providing comparisons to these baselines, addressing the concern about computational complexity, and introducing complexity results. In their Final Justifications, three reviewers acknowledge their appreciation for the authors' response and comment that it addresses their primary concerns.